# TopEC: prediction of Enzyme Commission classes by 3D graph neural networks and localized 3D protein descriptor

Karel van der Weg [1], Erinc Merdivan[2], Marie Piraud [2] & Holger Gohlke [1,3] ✉

Tools available for inferring enzyme function from general sequence, fold, or evolutionary information are generally successful. However, they can lead to misclassification if a deviation in local structural features influences the function. Here, we present TopEC, a 3D graph neural network based on a localized 3D descriptor to learn chemical reactions of enzymes from enzyme structures and predict Enzyme Commission (EC) classes. Using message-passing frameworks, we include distance and angle information to significantly improve the predictive performance for EC classification (F-score: 0.72) compared to regular 2D graph neural networks. We trained networks without fold bias that can classify enzyme structures for a vast functional space (>800 ECs). Our model is robust to uncertainties in binding site locations and similar functions in distinct binding sites. We observe that TopEC networks learn from an interplay between biochemical features and local shape-dependent features. TopEC is available as a repository on GitHub: https://github.com/IBG4-CBCLab/TopEC and https://doi.org/10.25838/d5p-66.

Proteins are at the basis of all cellular life. Since the first protein structure was elucidated in 1958, knowledge about the 3D structure has been instrumental in understanding molecular biology[1]. The shape of a protein occurs from specific interactions between atoms and their spatial relationship. These spatial relationships and chemical interactions give rise to the specific function of the protein. The Protein Data Bank[2] (PDB), the worldwide repository of experimental information about the 3D structure of biomolecules, contained 185,539 crystal structures at the end of 2021, of which 25,190 are non-redundant structures of enzymes, the focus of this study. Recent developments in protein structure prediction[3-5] massively improved the prediction of structural models of enzymes and led to the generation of large structural databases[6,7]. Yet, a predicted structural model is available only for 60% of all enzyme functions as proposed by the Nomenclature Committee of the International Union of Biochemistry and Molecular Biology (IUMBMB)[6]. Hence, accurately annotating molecular function to enzymes from (predicted) enzyme structures remains challenging. Determining enzyme function experimentally for many sequences is time-consuming, often enzyme function cannot be deduced directly from the structural representation, or the wrong enzyme function has been annotated to the sequence in databases[8]. Computational methods using enzyme structures as input can close this gap and allow for high-throughput enzyme function prediction.

Recently, deep learning and, especially, graph neural networks (GNNs) have gained in popularity and are used for a variety of chemical tasks, such as drug discovery[9], affinity prediction[10], protein interface prediction[11] and protein function prediction[12]. GNNs are a popular method for structural protein function prediction as most implementations are equivariant and translationally invariant[13]. On top of this, graphs closely relate to chemical representations[9]. However, representing a protein structure as a 3D graph is difficult, memory- and time-consuming[14]. To get around such limitations, often the graph is constructed containing only information from a projection of 3D to linear space[15,16]. Recent developments in GNNs allow us to encode the positions, distances, and angles between atoms in message-passing networks tested for small molecules[17-20]. In this study, we implemented

[1]Institute of Bio- and Geosciences (IBG-4: Bioinformatics), Forschungszentrum Jülich GmbH, 52425 Jülich, Germany. [2]Helmholtz AI Central Unit, Ingolstädter Landstraße 1, 85764 Oberschleißheim, Germany. [3]Institute for Pharmaceutical and Medicinal Chemistry, Heinrich Heine University Düsseldorf, 40225 Düsseldorf, Germany. ✉e-mail: h.gohlke@fz-juelich.de

the message-passing networks SchNet[18] and DimeNet++[20] for larger protein structures and used them for protein function classification (Fig. 1a, full details in "Methods").

Here, protein function relates to the specific reaction catalyzed by an enzyme as described by Enzyme Commission (EC) numbers[21]. EC numbers are represented by four hierarchical digits, specifying the main, sub-, and sub-subclass functions, as well as the specific enzyme function designation. For example, receptor protein-tyrosine kinase with EC number 2.7.10.1 is a transferase (2) transferring phosphorus-containing groups (2.7) as a protein-tyrosine kinase (2.7.10).

We intended to learn how to predict EC numbers from structure exploiting molecular recognition properties. While EC number prediction with machine learning is not new[22–25], relatively few structure-based methods explicitly encoding 3D information exist to predict EC numbers[12,26]. In EnzyNet[26], an abstraction of the protein to the 3D backbone information is used, and in DeepFRI[12], a graph is constructed from the 2D contact map created from the protein structure to reduce memory requirements compared to explicitly encoding 3D information. Furthermore, modern structure prediction methods, resulting, e.g., in the creation of the AF2 database[7], provided enough training samples to study structural enzyme function prediction at all four levels of classification.

Here, we present TopEC, a software package using GNNs for enzyme function prediction. TopEC can encode the 3D positions, distances, and angles between atoms and residues in graphs using a localized approach. We encode the atoms and residues following the atom type definitions in the force field ff19SB, allowing the network to learn different local chemical environments for one element. This allows us to investigate the differences between atom and residue-based enzyme function prediction (Fig. 1b) from a structural point of view. By encoding positional information and using predicted enzyme structures, we obtain an F1-score of 0.72 for predicting the EC designation when training on a fold split dataset using experimental and computationally generated structures. TopEC is trained without prior knowledge if a protein is an enzyme or not. This allows the user to scan proteins not classified as enzymes for enzymatic activity. For example, ABC transporters hydrolyze ATP[27] and G proteins hydrolyze GTP[28]. TopEC is a framework for rapidly devising, training, and testing deep-learning experiments based on protein structures. The software is written such that users can control the experiments from simple parameter files. Users can use TopEC to create their own function prediction tools or test different graph models with our pipeline for enzyme function prediction. TopEC is available as a repository, including accompanying data, on GitHub: https://github.com/IBG4-CBCLab/TopEC[29] and https://doi.org/10.25838/d5p-66.

## Results

### Overall strategy

Predicting EC numbers based on structural information without fold bias is challenging. While the fold of an enzyme generally determines function, even minor mutations can drastically alter function[30,31]. For example, TIM barrels and Rossman folds are groups with similar supersecondary structures but catalyze many different reactions[32,33]. If we solely took the overall shape (fold) into account, we would neglect the minor differences leading to different functions. We call this the fold bias. Typically, fold bias is removed by clustering the training,

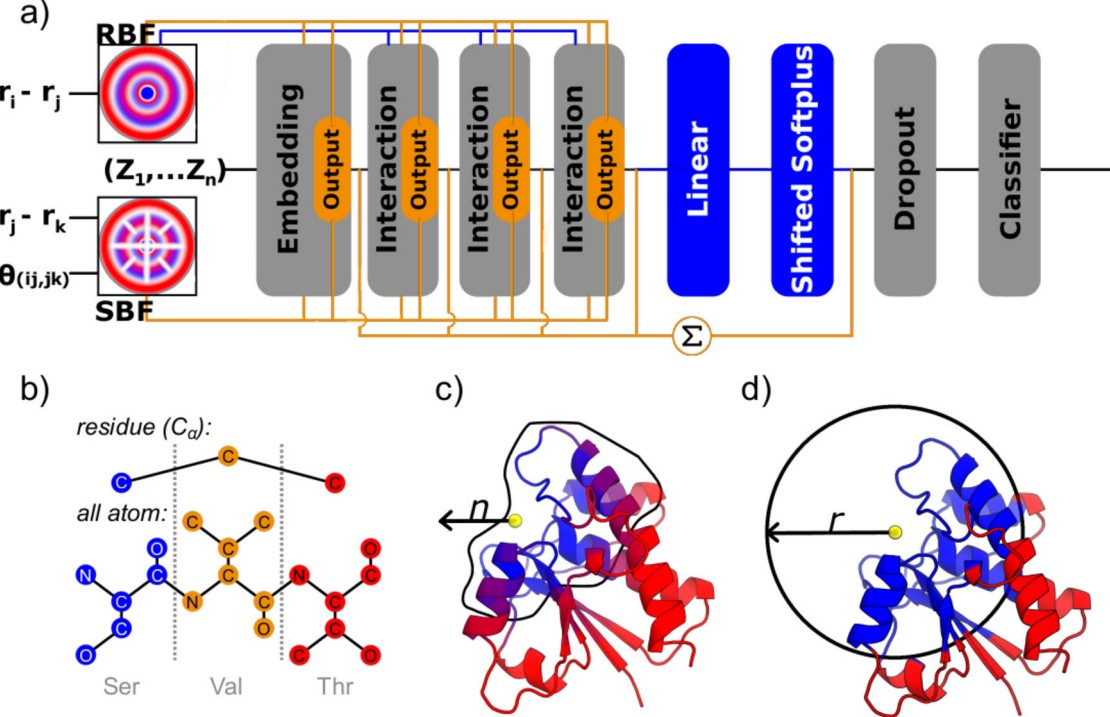

**Fig. 1 | Protein input to the neural network and network architecture. a** A schematic overview of the network architectures used. The orange path and blocks are unique to the DimeNet++ architecture, blue path and blocks are unique to the SchNet architecture, and gray blocks indicate commonality between both methods. In both methods, the atomic numbers $(Z_1,...,Z_n)$ are embedded similarly in the network. Radial Bessel filters (RBFs) are used to encode radial distances ($|r_i - r_j|$) between atoms, while Spherical Fourier Bessel (SBF) filters are used to encode distances ($|r_i - r_k|$) and angles $\theta_{(ij, jk)}$ between atoms $\{i, j, k\}$. Instead of a linear layer and shifted Softplus activation layer, DimeNet++ sums over all embedding and interaction blocks to generate the prediction. For more details on the network blocks see refs. 22, 24. A non-overlay version is available in Supplementary Fig. 1. **b** A schematic overview of the different resolutions tested in our localized 3D descriptor. In the *residue* view, all atoms besides the $C_\alpha$ positions are discarded and the nodes are encoded by amino acid type. In the *all-atom* view, each heavy atom is encoded as a node, and they are coded according to the chemical environment. The full scheme for atom encoding is available in the SI. Residues colored in blue are selected for the input, whereas those in red are discarded. **c** In the count representation, we expand from a point in space identified as a center of the binding site by the *n*-closest residues. **d** In the distance representation, we expand from a point in space and keep all residues within the radius *r*.

validation, and test splits by 30% sequence similarity[34,35]. Hence, we call our split "fold split". Deep neural network methods tend to perform poorly (F-score: 0.3–0.4) when fold bias is removed from the training dataset[12,26].

To solve both the issue of computational requirements and fold bias, we introduce a localized 3D descriptor based on the binding sites of enzymes. To identify the region where the chemical transformation likely takes place, we use experimental evidence, homology annotation, and the prediction method P2Rank[36]. From this region, we base the enzyme function classification on the closest $n$ atoms (Fig. 1c) or all atoms within a defined radius $r$ (Fig. 1d). We compare EC predictions at two resolutions of the enzyme structure (Fig. 1b). At residue resolution, we create a graph node for each $C_\alpha$ atom position of the enzyme backbone. At atom resolution, we create a graph node for each heavy atom position of the protein. The regional representations offer two advantages to a full structure representation: (1) We focus the network attention on learning the enzyme function from the binding site of the protein. (2) We reduce the GPU memory footprint as atomistic graphs of single enzymes do not fit on a NVIDIA A100 40 Gb GPU. We use the regional representations for the 3D-aware message-passing networks SchNet and DimeNet++ (Fig. 1a) to create TopEC-distances and TopEC-distances + angles. TopEC-distances is based on SchNet and used for creating localized 3D descriptors for atom and residue resolution. TopEC-distances + angles is based on DimeNet++ and used for creating localized 3D descriptors for residue resolution. We did not obtain satisfactory results for TopEC-distances + angles at atomic resolution due to memory constraints in the GPUs. The construction of atomic and residue resolution graphs is explained in the "Methods".

To evaluate the function prediction performance, we use the protein-centric F-score ($F_1$) as used for similar tasks in DeepFRI[12]. The balanced F-score measures accuracy as a function of the harmonic mean of precision and recall. The overall performance is averaged over all enzymes in the test set as in DeepFRI. All statistics are calculated using the PyCM[37] package. While we show a detailed breakdown below, we report only the highest performer over the parameter combination of network, size, and resolution in the Figures. Complete statistical reports with confusion matrixes are available in Supplementary Data 1 for each combination tested.

## Datasets used

Due to the high EC redundancy in the PDB[38] and the resulting low amount of diverse enzyme functions with experimentally known binding sites, we trained the networks on a combination of datasets with experimental and predicted protein structures. We now describe the datasets (Table 1) that are contained in the TopEnzyme[6] database we released earlier.

First, we used experimental structures. For this, we filtered the Binding MOAD[39] for enzyme structures with an associated EC and binding interface. This results in 21,333 experimentally determined enzymes covering 1625 different enzyme functions. We call this filtered set the Binding MOAD dataset. To supplement it, we generated homology models with TopModel. This yielded 8904 predicted enzyme structures covering 2416 enzyme functions. This is the initial TopEnzyme dataset. We chose this approach as it provides experimentally determined binding sites from the Binding MOAD and accurately derived binding sites inferred from homologous templates used for structure prediction. We tested two data splits on Binding MOAD and TopEnzyme: the temporal split and the fold split as in ECPred[23]. For the fold split, we used MMseqs2 to cluster our database at 30% sequence identity. All data sets are split over training, validation, and test sets in approximately 80%/10%/10% ratios. We also tested the combination of Binding MOAD and TopEnzyme and called this the Combined dataset.

To gain access to a larger and more diverse structure database for training, we supplemented the experimental dataset with predicted binding site information. We filtered the PDB keeping only EC designations with at least 50 structures using a fold split, for a total of 300 enzyme classes across 56,058 structures, resulting in the PDB300 dataset. The binding sites are obtained with P2Rank when we lack any experimental information. Again, we use MMseqs2 to cluster our database by 30% sequence identity with approximately 80%/10%/10% training, validation, and test set ratios.

With the release of the AF2 database[7] (AF2 DB), we further supplemented the computational dataset with enzyme structures generated by the end-to-end method AF2. To fully utilize predicted enzyme structures, we expanded TopEnzyme by incorporating AF2 structures available from the AF2 DB and released it as a separate database[6]. We filtered the UniprotKB/Swiss-Prot database for proteins with an associated EC. After binding site prediction with P2Rank, we removed all enzymes without a predicted binding site, yielding 201,107 enzymes covering 703 unique ECs with at least 50 different computational structures, resulting in the AF703 dataset.

To obtain biochemical insights into the performance of the network models, we introduce two benchmark datasets. The first is a modified Price dataset. The Price[40] dataset refers to instances where the SEED[41] or KEGG[42] databases had prior instances of either misannotation or inconsistent annotation[43] based on predictions derived from sequence information. Second, the ProSPECCTs dataset is divided into ten different (sub)categories (DS1-DS7, subcategories: DS1.2, DS5.2, DS6.2) benchmarking various properties of enzyme binding sites[44]. Details of the content of each dataset are available in Supplementary Table 1. After removing ECs we did not train with the AF703 and PDB300 datasets, we are left with 5543 structures. These classes were removed from training for not containing at least 50 samples.

Finally, we adapted the Catalytic Site Atlas (CSA) dataset along with all binding residues from BioLiP[45]. Again, we removed all ECs we did not train with the AF703 dataset. We did not test the CSA with networks trained on the PDB300. As the CSA exists as a database within the PDB, there would not be enough training examples left after properly separating the data into train and test sets. This gave 364 enzymes with experimental structures spanning 295 ECs for which we have the full binding information. The filtered Price, ProSPECCTs, and CSA datasets are reported in Supplementary Data 2.

## Local information from the binding site is sufficient to classify Enzyme Commission numbers

Initially, we tested the prediction performance for the seven main classes of enzyme functions according to EC using the localized 3D distance descriptor. We trained the networks with oversampling to reduce bias for the more populated main classes (Supplementary Fig. 2). For comparison, we re-trained EnzyNet, a 3D-CNN, and Deep-FRI, a GCNN, on the data sets using the best settings as described in the respective publications. Additionally, we also re-trained EnzyNet and DeepFRI on the localized descriptor with a radius cut-off of 16 Å around the binding site. The results are shown in Table 2A. As EnzyNet returns results for multiple modes, we show the full results in Supplementary Fig. 3.

In five of the six tests, TopEC-distances + angles at residue resolution outperforms all other neural networks with an average increase of the F-score of 0.04 compared to TopEC-distances and, more markedly, 0.13 to EnzyNet and DeepFRI. Only on the combined dataset for the temporal split, TopEC-distances at residue resolution outperforms the rest. Generally, the topology (fold) of an enzyme is thought to be the major determinant of the given function[30,31]. This leads to higher performance on the temporal split as many identical topologies are present in both the training and test sets. In the case of the fold split, the data is separated by 30% sequence identity with no common folds in either the training, validation, or test sets. Thus, the network cannot learn a function from topology, which reduces the bias for proteins with a similar topology but a different function. This ability is

**Table 1 | Description of the datasets used**

| Dataset | Proteins (n) | ECs (n) | Application |
|---|---|---|---|
| Benchmark datasets: | | | The training, validation, and test set are split according to a 30% sequence identity. Data is divided into roughly 80%/10%/10% ratios. |
| Binding MOAD | 21,333 | 1625 | Binding MOAD consists of proteins and experimentally determined binding sites. This is the original input to test if a binding site is sufficient for classification. |
| TopEnzyme | 8904 | 2416 | Created using TopModel to supplement Binding MOAD. Also used to compare homology models against ab initio folded models from AlphaFold2 in Supplementary Fig. 4. |
| Combined | 30,237 | 3047 | Combination of Binding MOAD and TopEnzyme to obtain more training samples. |
| PDB300 | 56,058 | 300 | Created using experimental enzyme structures in the PDB. |
| AF703 | 201,107 | 703 | Created using all UniProtAC identifiers present in Swiss-Prot with annotated enzymatic activity. The computational enzyme structures are then obtained from the AlphaFold2 DB using the same identifier. |
| Investigative datasets: | | | The datasets have been filtered for proteins with enzymatic activity only. Training and validation splits are based on a combination of AF703 and PDB300, removing all enzymes with >30% sequence identity to the investigative dataset. |
| Price | 22 | 12 | Previously mis-annotated enzymes in SEED[34] or KEGG[35]. |
| PrOSSPECTs | 3968 | 280 | Benchmark dataset specifically designed to test a variety of properties for protein binding site prediction. |
| CSA | 364 | 295 | Catalytic site atlas containing annotations for each residue participating in the catalytic mechanism and its role. |

n refers to the number of items.

**Table 2 | Prediction performance (*F*-score) for the networks tested on multiple data sets[a]**

**(A) Main class classification**

| Dataset | Networks | | | | | | |
|---|---|---|---|---|---|---|---|
| | TopEC-distances (atoms) | TopEC-distances (residues) | TopEC-distances + angles (residues) | EnzyNet | EnzyNet(local) | DeepFRI | DeepFRI (local) |
| Temporal split: | | | | | | | |
| Binding MOAD | 0.73 | 0.73 | **0.75** | 0.71 | 0.63 | 0.57 | 0.58 |
| TopEnzyme | 0.52 | 0.49 | **0.53** | 0.41 | 0.42 | 0.40 | 0.41 |
| Combined | 0.75 | **0.76** | 0.73 | 0.52 | 0.53 | 0.54 | 0.56 |
| Fold split: | | | | | | | |
| Binding MOAD | 0.60 | 0.62 | **0.66** | 0.52 | 0.55 | 0.59 | 0.57 |
| TopEnzyme | 0.45 | 0.44 | **0.51** | 0.37 | 0.44 | 0.36 | 0.35 |
| Combined | 0.62 | 0.60 | **0.63** | 0.52 | 0.53 | 0.54 | 0.56 |

**(B) Hierarchical class classification**

| Dataset: Combined | Networks | | | | | |
|---|---|---|---|---|---|---|
| | TopEC-distances | TopEC-distances + angles | EnzyNet | EnzyNet (local) | DeepFRI | DeepFRI (local) |
| Temporal split | 0.75 | **0.76** | 0.73 | 0.65 | 0.43 | 0.61 |
| Fold split | **0.57** | 0.55 | 0.51 | 0.53 | 0.20 | 0.25 |

**(C) Hierarchical class classification with experimental or predicted binding sites**

| Binding site origin[b] | Network | |
|---|---|---|
| | TopEC-distances | TopEC-distances + angles |
| Binding MOAD | **0.57** | 0.56 |
| P2Rank | **0.59** | 0.53 |

[a]The best-performing networks are highlighted in bold.
[b]These are the same structures as for Binding MOAD in Table 2A, except that the center location of the binding site for the localized 3D descriptor is determined by P2Rank instead of experimental evidence.

important for applying enzyme function classification in enzyme discovery or engineering synthetic enzyme cascades[46].

The performance on the TopEnzyme dataset is markedly lower than on the Binding MOAD dataset. The TopEnzyme section of comparative models likely does not contain enough samples to learn local chemistry from the training set: In this TopEnzyme section, there are on average 2–3 training models for each test model as to a specific EC number. Thus, the TopEnzyme section contains a large variety of chemistry at the local level, but only a few examples for each case. This is likely also the reason why the performance on the combined dataset is lower than on the Binding MOAD dataset, even though combinations of experimental and predicted structures have been shown to improve results[12].

In EnzyNet, the enzyme information is reduced to a topological view. It performs better on the Binding MOAD dataset with a temporal split than on datasets with less topology overlap. DeepFRI, where enzyme information is created from a 2D contact map, generally performs the worst. Our local descriptor performs better than EnzyNet and DeepFRI for all datasets, likely because it implicitly directs the networks into the more relevant regions for enzyme function. Only on the temporal dataset, EnzyNet outperforms the local representation, as here it can learn from the topological view.

Next, we evaluated the performance of the networks when predicting the full hierarchy of EC numbers using the combined dataset. We show the results for TopEC-distances and TopEC-distances + angles against DeepFRI and EnzyNet with the local descriptor in Table 2B. For the temporal data split, TopEC-distances + angles performed the best, while for the fold data split, TopEC-distances performed the best, with an average increase of 0.06 and 0.29 of the *F*-scores compared to EnzyNet and DeepFRI, respectively. We also tested the influence of the graph count and location of the localized 3D descriptor on the performance (Supplementary Fig. 4). The

performance starts to plateau around 75–100 nodes per enzyme graph depending on the model. TopEC-distances + angles performs worse with larger graphs as we start to encounter out-of-memory issues on the used GPUs. When we choose random centers for defining a binding site for the localized 3D descriptor, the models perform worse. For the hierarchical classification, as for the main classification, representing the local chemistry well to steer the network implicitly into the relevant region is likely more important than the overall topology-function relationship. In other full-structure methods, saliency maps of the network overlap with experimental binding sites[12,47]. In TopEC, structural information is encoded explicitly compared to EnzyNet and DeepFRI (see above). Our results indicate that encoding structural information explicitly is important when classifying enzyme functions without considering topology-function relationships.

Finally, we trained our networks using experimental binding sites from the Binding MOAD or binding sites predicted by P2Rank[36] on experimental structures with a fold split (Table 2C). Although binding site predictors are not perfect, some structural noise could make the machine learning method more robust to uncertain information[48–50]. The predictive performance is similar for both of our networks and both binding site origins. Hence, we will use P2Rank to obtain binding site information in larger databases when experimental information is lacking.

## Expanding the chemical space with computationally generated enzyme structures improves the predictive performance of the networks

Before we test for computationally generated enzyme structures, we want to obtain a baseline performance. We created a fold split for experimentally determined enzyme structures using predicted binding sites for more training samples, the PDB300 dataset. The area under the precision-recall curve (AUPR) for each EC is shown as a distribution in Fig. 2a. The *F*-score for this dataset is low with 0.66 for

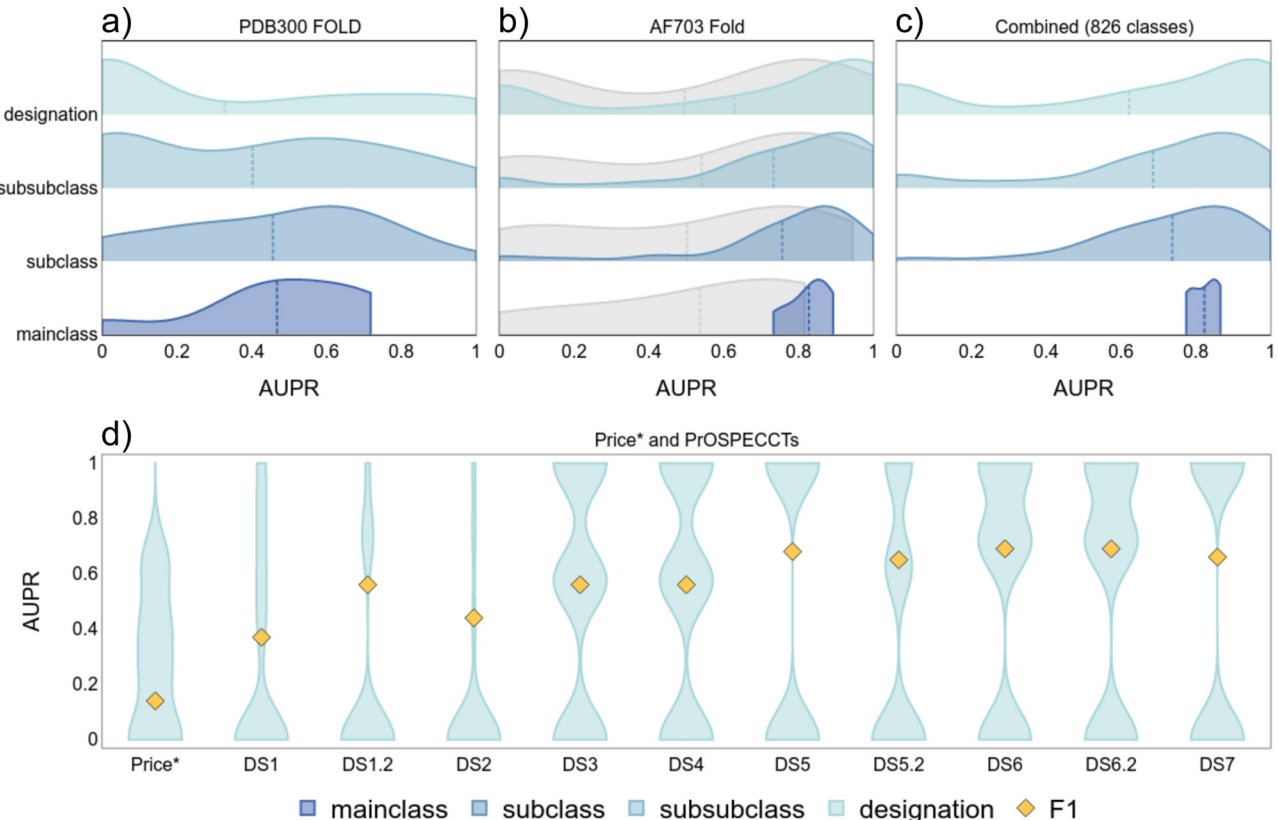

**Fig. 2 | The predictive performance of TopEC networks is shown by the distribution of AUPR for each EC class at residue resolution. a–c** The dotted line represents the average AUPR score over all proteins for the TopEC networks. We show the distribution of AUPR for each hierarchy of EC numbers. **a** The predictive performance for the best TopEC network trained on the PDB300 dataset using a fold split. **b** Similar to **a**, but we used the AF703 dataset. The gray distribution represents the best network from Table 2. **c** Similar to **b**, but we combined the PDB300 and AF703 datasets to yield 826 EC classes. **d** The predictive performance for the best fold-split TopEC network trained on AF2 structures; only the full designation is displayed. This is a reduced version of the Price and ProSPECCTs dataset as not every entry has an annotated EC present. For a description of each dataset, see the "Dataset" section. The yellow diamond indicates the *F*1 score. Source data are provided as a Source Data file.

main-, 0.53 for sub-, 0.45 for sub-subclass, and 0.39 for the designation level.

There are a few issues we encountered when training with the PDB300. Many classes are poorly predicted as we have limited samples. For example, for translocases (*F*-score: 0), we only have 228 experimental structures for training, validation, and testing. Furthermore, while experimental evidence is regarded as the ground truth when creating prediction models, the PDB has known issues such as redundancy[51] or enzyme-inhibitor complexes marked as enzyme-substrate complexes[52–54]. For serine endopeptidase La (EC: 3.4.21.53), none of the eight test PDBs are correctly predicted. Notably, one is described to have no activity (4fwh), three are characterized in complex with inhibitors (4fw9, 4fwd, 4fwg), and the other four are from a mutation study where they mutated the catalytic dyad residues (7ev4, 7euy, 7ev6, 7eux)[53,54]. Yet, these structures are all marked as having enzymatic activity with EC classification 3.4.21.53 in the PDB.

Next, we trained our networks with structural models generated by homology modeling from TopEnzyme or structures from the AF2 database (AF2 DB) for identical Uniprot IDs (Supplementary Fig. 5, 7924 structures). For the networks trained at atomic resolution, AF2 structures generally increase the predictive performance (*t*-test, $n = 7.924$: *p* value = 0.041). For networks trained at residue resolution, performances are more similar. The results indicate that comparative models and models generated by an end-to-end method perform equally in TopEC. An advantage of homology models is that we can utilize crystal structure homologs to refine the binding site prediction.

This procedure is described in the "Methods" subsection "Binding sites".

The performance of the fold-split network on AF2 structures is shown in Fig. 2b and compared to the best fold-split network obtained with the combined dataset of experimental structures and comparative models (Table 2B). Using the AF2 dataset majorly improves the *F*-score on the main (+0.18), sub (+0.28), sub-subclass (+0.33), and designation (+0.33) levels. Also, the number of designations spanning the seven main classes increases from 97 to 703. This leads to a major AUPR increase for underrepresented classes such as Lyases (+69 new designations, +0.28), Ligases (+56 new designations, +0.89), and Translocases (+21 new designations, +0.64). A similar distribution shift to higher AUPR values is obtained for sub, sub-sub, and hierarchical classes in Fig. 2b. While the classification results improve as we decrease the specificity of the enzymatic function, i.e., move from the designation to the main class, there is no correlation between the number of structures, folds, pLDDT-score, or designations and the performance (Supplementary Fig. 6).

We also tested a random data split using the AF703 dataset. Despite most classes being perfectly predicted due to the high overlap of training and test data, the network was not able to predict any type II site-specific deoxyribonucleases. While the network is likely biased toward the overall fold, the folds for this EC are highly divergent (Supplementary Fig. 7). Interestingly, the fold-split network correctly predicted half of them, indicating that information can be gained using the local descriptor if fold bias is reduced.

Finally, we tested the performance of the network using the combined AF703 and PDB300 datasets with a fold split for a total of 257,165 structures covering 826 EC designations. The overall performance is slightly worse compared to the network trained on solely AF2 models (main class: ±0.00, subclass: −0.02, sub-subclass: −0.02, designation: −0.01). However, by combining the datasets, we can predict a larger variety of classes (826 vs. 703) and improve the prediction performance on experimental structures. We reached an $F1$-score of 0.72 for the combined dataset at the designation level. We tested this dataset on EnzyNet and CLEAN[55] and obtained $F1$-scores of 0.50 and 0.74, respectively. We were not able to obtain satisfying results for DeepFRI as this method optimizes for both positive and negative probabilities, which tends to overfit on the negative probabilities in the case of a large number of classes. Compared to the network trained on PDB300, the $F$-score increases on experimental structures for main classes (+0.09), subclasses (+0.14), sub-subclasses (+0.17), and designations (+0.28) when using the combined datasets. To further evaluate the impact of the fold bias on the network's performance, we generated an additional test set using Foldseek clustering. For initially comparing the Foldseek clustering to the MMSeqs2-based clusters, the Jaccard similarity was calculated for each enzyme pair present in the clusters. The similarity is 0.43, indicating how the clusters formed by Foldseek are distinct from the MMSeqs2 clusters. After the removal of all Foldseek clusters with overlap with the training and validation sets from the original test set (Fig. 2), the $F$-score is 0.69 for the main class, 0.61 for the subclass, 0.57 for the sub-subclass, and 0.52 for the designation classification. Note that the network was not retrained on a training and validation set following Foldseek criteria but the performance assessment was limited to the enzymes that are clustered according to Foldseek. The result suggests that a 30% sequence identity cut-off may not fully eliminate fold bias.

## Benchmarking the price and ProSPECCTs datasets

We tested the performance of the AF2 fold-split network on the modified Price[40] and ProSPECCTs[44] datasets (Fig. 2d). The results show the AUPR curve for the designation level of EC with the $F_1$ score overlaid. While the average AUPR for the enzymes in Price are low, the network is more confident in the correct predictions (0.87 confidence score) than in incorrect predictions (0.56 confidence score). Interestingly, the confidence score is still high for cases where the first three digits of EC are correct (0.71 confidence score) compared to cases where 2 or more digits are incorrect (0.44 confidence score). These datasets are designed to be difficult, i.e., the availability of structural enzyme data for these ECs from other folds is low, leading to lower results due to limited training samples. Enzymes of the Price dataset are less conserved, and many of the enzymes have been previously misclassified by in silico methods[40]. Similarly to ProteInfer, we try to recover the new corrected classifications. Compared to ProteInfer, we predict more enzymes correctly, but also more enzymes wrongly, as ProteInfer contains a mechanism to not make uncertain predictions.

Generally, for ProSPECCTs, we predict an enzyme class either completely correct or wrong, leading to the hourglass-shaped AUPR curves. For the wrongly predicted classes, we lack the diversity in ECs from distinct folds after removing any training samples with >30% sequence similarity to ProSPECCTs.

For the ProSPECCTs dataset, we test the performance on structures with identical sequences (DS1) and containing similar ligands (DS1.2) (Fig. 2d). Although the overall performance shown in Fig. 2d is only fair, the performance on structures with identical sequences (DS1) and similar ligands (DS1.2) should be more akin to DS5 and DS6. Only for one structural group, the wrong EC was predicted. This group contained significantly more samples, skewing the results. This could be due to the EC number not being a good representation of the function in this case. The network predicted all non-chaperonin molecular chaperone ATPases (EC: 3.6.4.10) as histidine kinase (EC: 2.7.13.3); however, by now, 9 of the 13 unique designations of sub-subclass 3.6.4 have been

moved to the Isomerase main class, showing how new understanding can lead to changes EC assignment. Thus, the EC number of an enzyme is not always the best representation of function and changes with better understanding, leading to conflicting information in databases.

The poor performance for DS2 likely arises from this dataset focusing on flexible NMR structures. Although we train the network with some translational noise, this is different from enzyme movements captured in NMR ensembles. As we did not train on NMR ensembles, the network did not learn about such movements. Next, we tested a set of structures with different physicochemical properties (DS3) and different shape properties (DS4) of the binding sites. The performance is low indicating that our networks do not understand changes in local chemistry and shape if they have not seen these changes before. In DS5 and DS5.2 (including phosphate binding sites), we test the classification of proteins that bind to identical ligands and cofactors. As we trained without the presence of ligands, we expect the network to perform well here even without any ligand bias. In DS6, we test a set of structures with distant relationships between protein binding sites but identical ligands that have a similar environment; DS6.2 additionally includes cofactors. In DS7, the recovery of known binding site similarities within a diverse set of proteins is tested. The networks perform well on DS5-7 ($F1$ score >0.6), which all describe similar enzyme functions despite distinct binding site environments, indicating that the networks are robust with respect to function prediction in the context of a chemical environment. A full overview of the performance on the Price and ProSPECCTs datasets for all tested network types is given in Supplementary Fig. 8.

## The network learns from an interplay of chemical interactions and local shapes

Since we decoupled the overall shape from the enzyme function by using a fold split along with the localized 3D descriptor, the network learns from local shape motifs. To scrutinize this, we analyzed our networks using a modified version of GNNExplainer[56], a model-agnostic tool for investigating graph neural networks. As we already identify the subgraph structure best describing the enzyme function (binding pocket), we replace the subgraph identification with a softmask method adapted from GNNExplainer, since we only aim to qualitatively describe residue or atom importance. The softmask method is expected to learn the relative importance of each node to the prediction. First, we applied GNNExplainer to the TopEC network trained on AlphaFold2 structures with a fold split using only $C_\alpha$ positions of residues and the CSA dataset. Thereby, we correctly predict 232 functions for all four EC hierarchies. The correctly predicted subset is used for GNNExplainer to obtain the node importance, which is normalized and mapped onto each residue, providing an overview of importance per residue (Fig. 3a and Supplementary Fig. 9). For aspartic acids, glycines, and threonines, binding site and catalytic residues are identified to be more important. Yet, important residues are not limited to the binding site, as seen for cysteine, phenylalanine, proline, and tryptophan. Glycine is the most important residue across all quartiles. Since we only consider the $C_\alpha$ positions, glycine is the chemically most completely described residue in the network, whereas for other residues side chain positioning may be critical.

To test the influence of chemical information arising from the local shape or expert chemical knowledge, we used the 364 enzymes obtained from the CSA. The distribution of AUPR for each class is shown in Fig. 3b. The distribution is broad at zero and one, because there is often only one enzyme for a specific class in this dataset (295 classes over 364 enzymes). To test the influence of chemistry versus shape, we computationally mutated all residues to alanine, which does not alter the protein shape for the residue networks (Fig. 3c). We also mutated only the catalytic residues to alanine (Fig. 3d). In both cases, the network performance drops markedly. We tested how many residues we can randomly mutate before we lose chemical knowledge for accurate predictions (Fig. 3e). While the average AUPR slowly

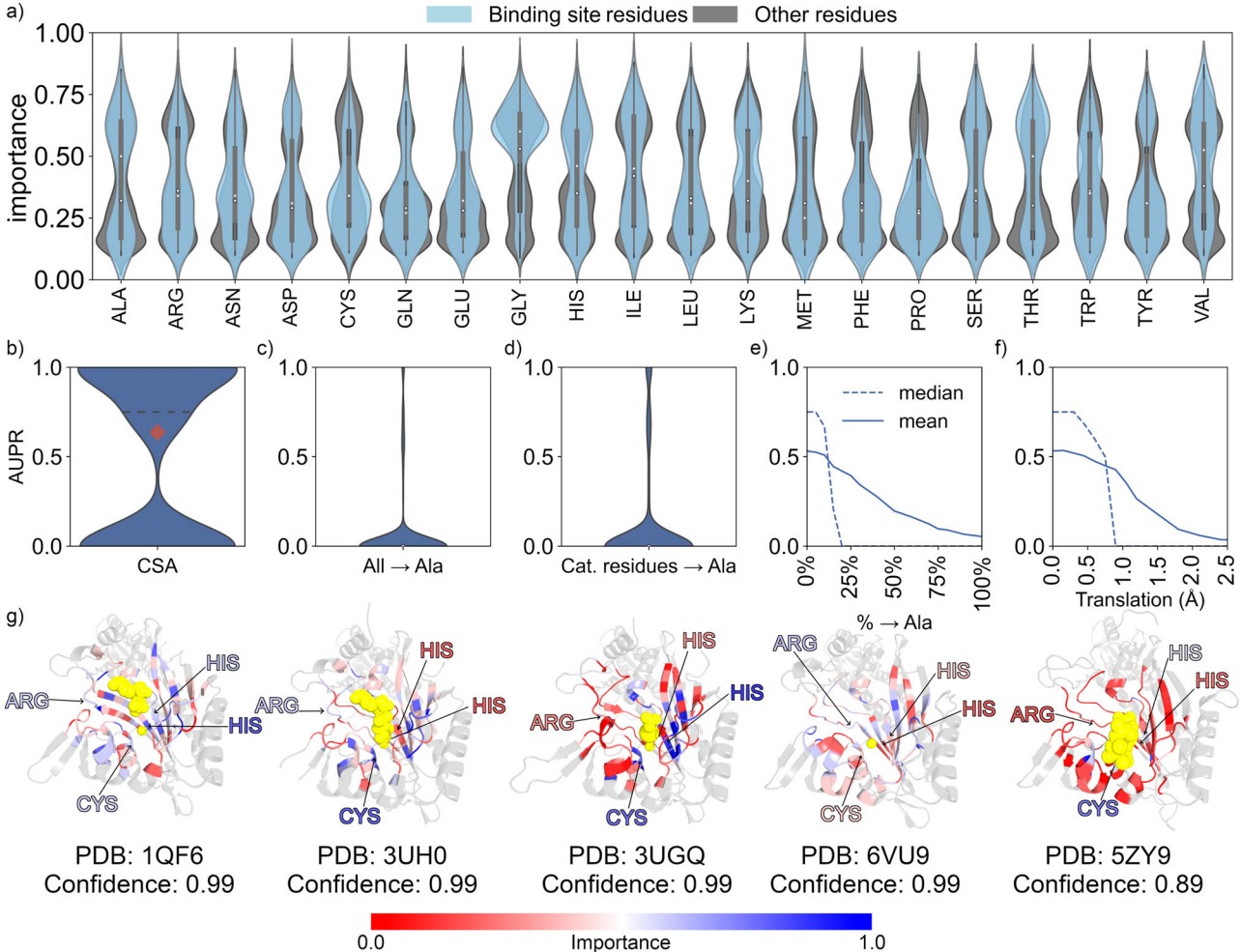

**Fig. 3 | The importance of network nodes for the prediction at residue resolution as determined with GNNExplainer.** Each network node corresponds to the $C_\alpha$ position of a residue. All structures are obtained from the PDB, and the residue positions for catalytic sites are obtained from the catalytic site atlas (CSA). Binding residue positions are obtained from the BioLiP database. All structures are predicted with the network trained on AF703 with a fold split using $C_\alpha$ positions for each residue. PDB structures with ECs not seen in training are removed from the dataset. **a** Importance of each network node for catalytic and binding residues (blue) and other residues (gray) within the localized 3D descriptor ($n$ = 232 catalytic sites). The black bars represent the first and third quartiles, while the white dot represents the median value. **b** AUPR for all tested classes. The red diamond shows the macro-averaged (unweighted mean) $F1$ score. The dotted line shows the median

score. **c** Similar to **b**, except all residues are mutated to alanine in the network. **d** Similar to **b**, except all catalytic residues are mutated to alanine in the network. **e** The mean and median AUPR as a function of the percentage of mutated residues to alanine. The mutated residues are selected randomly. **f** The mean and median AUPR as a function of random translation of atom positions. **g** Five correctly predicted threonine-tRNA ligases (EC: 6.1.1.3). The gray-colored sections are removed from the protein when creating the localized 3D descriptor. The ligands are shown in yellow. The color scale shows the importance as predicted by GNNExplainer. The catalytic residues are pointed to and colored by importance. The confidence indicates how certain our network is of the prediction. Source data are provided as a Source Data file.

decreases with more mutations, the median AUPR plummets as soon as we mutate ~20% of the residues, indicating that local side chain information is critical to the learning process. To test how much we can disturb the shape representation, we randomly translated residue positions from 0.0 to 2.5 Å (Fig. 3f). Translations starting from 0.75 Å decrease network performance, which corresponds to a resolution of a crystal structure of ~4.5 Å[57].

For five correctly predicted threonine-tRNA ligase domains from different PDB structures, catalytic residues are colored according to the normalized importance from GNNExplainer (Fig. 3g, larger size available in Supplementary Figs. 10–14). While the structures have the same fold, the sequence similarity is ~35–60% over these five structures, except between 3UH0 and 3UGQ (100%, Supplementary Table 2). In all five cases, residues on the colored β-strands are alternatingly important although the importance does not correspond to the side chain orientation towards the substrate or not. The importance difference does increase with increasing distance to the

substrate, pointing to the relevance of local chemical information. When testing residue stability using Luque[58], Thermometer[59], Constraint Network Analysis[60], the Amino Acid Interactions webserver[61], or K-Fold[62], we found no correlation between predicted importance and residue stability (Supplementary Figs. 15–21). We have not checked for conservation or connectivity as a reason for importance.

Overall, disrupting either local chemistry or shape decreases network performance. In turn, network performance likely results from an interplay between biochemical features, such as specific interactions or conserved motifs, and local shape, as well as atom type-dependent features that are more difficult to reconcile with general protein biochemistry understanding.

## Classifying at the atom resolution improves the understandability of the neural networks

When using the Cα position in the local descriptor, information on the position of side chain atoms is lost, as we only encode the Cα position

of the backbone labeled with the amino acid type. Our analysis revealed that specific Cα backbone positions were identified as critical (Fig. 3g). However, these positions appeared to alternate without a clear pattern. The alternation does not coincide with the direction of residues in α-helices and β-sheets that point toward the binding site. We could not link the identified positions to known chemical or biological expertise. This suggests for the Cα position in the local descriptor that the model may rely on structural features not easily interpretable with established biochemical knowledge. Alternatively, the patterns might result from GNNExplainer highlighting noisy patterns that arise from the complex interplay of graph features in the network.

To investigate this further, we adapted GNNExplainer to investigate a TopEC network trained on AlphaFold2 structures with a fold split and the closest 150 heavy atoms to the binding center with the same CSA dataset as in the previous section. The performance decreases from 232 to 142 correct predictions when using the higher-resolution atomistic descriptor. This drop likely reflects the increased complexity of the graph representation, which could obscure the signal for enzymatic function already encoded in the local 3D position of the amino acid chain. For example, CLEAN achieves impressive performance using only sequence information and the ESM base model. Adding more atoms to the graph may introduce noise, making it harder for the model to isolate functionally relevant features. We tested the influence of the small binding site representation by predicting these 90 incorrect cases with residue resolution limited to the positions in the atom resolution. For all 90, we did not obtain correct predictions with this residue resolution model.

However, we do obtain interesting insights into the network's local properties. For correctly predicted enzymes, we found that biologically relevant atoms, such as those participating in educt stabilization and the catalytic reaction tended to have high importance values. We exemplified this for three serine endopeptidases at residue (Fig. 4a–c: left) and atomic resolution (Fig. 4a–c: right), repressor LexA[63], type 1 signal peptidase[64], and GlpG[65].

For repressor LexA (Fig. 4a) at residue resolution, residues around the binding site are more important for the prediction. At atomic resolution, Gly117 is of low importance, except for the backbone interaction together with Asp 127 to form the oxyanion hole[63]. The backbone and $O_\gamma$ atoms in Ser119 together with Lys159 are important for the prediction. Interestingly, the $N_\zeta$ position of Lys159 is less important than the hydrophobic side chain, which can result from the hydrophobic packing interactions of Met120 and Ile179 as described in ref. 63. Conversely, the $S_\gamma$ and $C_\varepsilon$ positions of Met120 that pack against the Lys159 side chain are considered important. By contrast, for Ile179, the importance is low. One reason could be that the network did not see the full side chain as some atoms were not part of the localized 3D descriptor due to the cutoff or that the information is contained in the neighboring Lys159 embeddings.

For type 1 signal peptidase (Fig. 4b), the catalytic Lys145 is the most important residue, and its $N_\zeta$ atom is the most important atom at atom resolution. At atomic resolution, multiple serine residues with importance values related to functional relevance are in an interaction range to Lys145: Ser90 initiates the nucleophilic attack on the ligand with its $O_\gamma$[64], which is found to be most important; the $O_\gamma$ of Ser278 is within range for hydrogen bonding with the $N_\zeta$ of Lys145[64], although only $C_\beta$ is found to be important; Ser88 is involved in oxyanion hole formation[64], which is concordant with the Ser88 side chain being marked as unimportant. Phe133 is found to be highly important and is in van der Waals contact with the Lys145 side chain. The Tyr143 $C_\gamma$, $C_\delta$, and $C_\varepsilon$ atoms are also in van der Waals contact, with the first two atoms being more important.

The rhomboid protease GlpG (Fig. 4c) is an interesting case, as at residue level it is predicted as a translocase but at atom resolution it is classified as a serine endopeptidase. This misclassification

highlights the importance of decoupling the overall fold from the catalytic interactions, as this serine endopeptidase is found within the membrane but is not a translocase[65]. The Ser201 $O_\gamma$ interacts with His254 through a strong hydrogen bond[65] characterized as the most important interaction by the network. His254 also stacks on Tyr205, which is considered important for the function of the dyad[65]. Gly199 is found to be important, too, and contributes to the oxyanion binding[65]. Asn154 is positioned too far away to form a catalytic triad instead of a dyad[65]; its $O_{\delta1}$ and $N_{\delta2}$ atoms are classified as less important, although the rest of the side chain is found to be important. Interestingly, depending on the context of the reaction, this asparagine can participate in catalysis, which can explain the importance of the nonpolar side chain atoms. Asn154 is also in the interaction range with His145[65], indicating why His145 is classified as more important.

One limiting aspect of the importance explanation is separating atoms participating in catalysis and binding from the rest of the localized 3D descriptor. The importance range for non-catalytic and -binding atoms is broadly distributed (Supplementary Fig. 23). Still, the high importance of catalytic and interacting atoms agrees with expert knowledge. To understand this finding, we looked at the importance distribution for the most common catalytic residues in the CSA (Fig. 4d). We show how many atoms we have seen for every explained residue type. We grouped and colored the residues by type, charged (blue), polar (red), aromatic (yellow), and hydrophobic (purple). The residues in the last group are usually not associated with catalysis, however, they can serve as scaffolding residues in some catalytic reactions. Hence, they may be marked as important for catalysis in the CSA. The distribution for the other catalytic residues is shown in Supplementary Fig. 23. Although the importance distribution varies considerably, the network lays more importance on atoms participating in catalytic reactions. For histidine, importance values are higher for the $C_{\varepsilon1}$, $N_{\delta1}$, and $N_{\varepsilon2}$ atoms; for aspartic acid, this is so for the $O_{\delta1}$ and $O_{\delta2}$ atoms and for glutamic acid for $O_{\varepsilon1}$ atoms; unexpectedly, $O_{\varepsilon2}$ of glutamic acid is less important. The difference between $N_{\delta1}$ and $N_{\varepsilon2}$ atoms in histidine could arise from its different positions in the imidazolyl ring and the respective role in enzyme function[66–68]. In arginine, the $C_\zeta$ position is more important than the $N_{\eta1}$ or $N_{\eta2}$ atoms, which might indicate that the $C_\zeta$ embedding is updated from the location of the $N_{\eta1}$ and $N_{\eta2}$ atoms. Furthermore, the $C_\zeta$ atom often participates in π-stacking interactions[69]. For lysine, besides the $N_\zeta$ atoms, the $C_\alpha$, $C_\beta$, $C_\delta$ atoms are important, which can participate in hydrophobic interactions[70]; $C_\varepsilon$ and $C_\gamma$ also participate in hydrophobic packing, although found not to be important by the network. This could be because the position of the side chain, and thus the hydrophobic packing, is already largely defined by the positions of the $C_\beta$ and $C_\delta$ atoms. In tyrosine, the $C_\zeta$ and $O_\eta$ atoms are the most important ones; $O_\eta$ can be involved in hydrogen bonding and change the protonation state. For serine, we expected to see more importance on $S_\gamma$ and $O_\gamma$ atoms. The absence might be explained due to the small residue size, as other atom positions are close enough to share embedding updates.

We also obtained the importance for each catalytic and binding residue for all cases wrongly predicted at atomistic resolution (Supplementary Fig. 24). The importance of side-chain atoms is generally higher than for backbone atoms. However, the importance is less prominent for the interacting atoms than in the correctly predicted results. Using the importance results for a refinement step with a learning objective of matching the importance of atoms participating in catalysis according to experiment may improve the predictive performance but this remains to be tested. This might lead to a closer match of importance prediction and function prediction performance, which might be useful for finding and designing new enzymes and deciphering how enzyme function prediction networks learn.

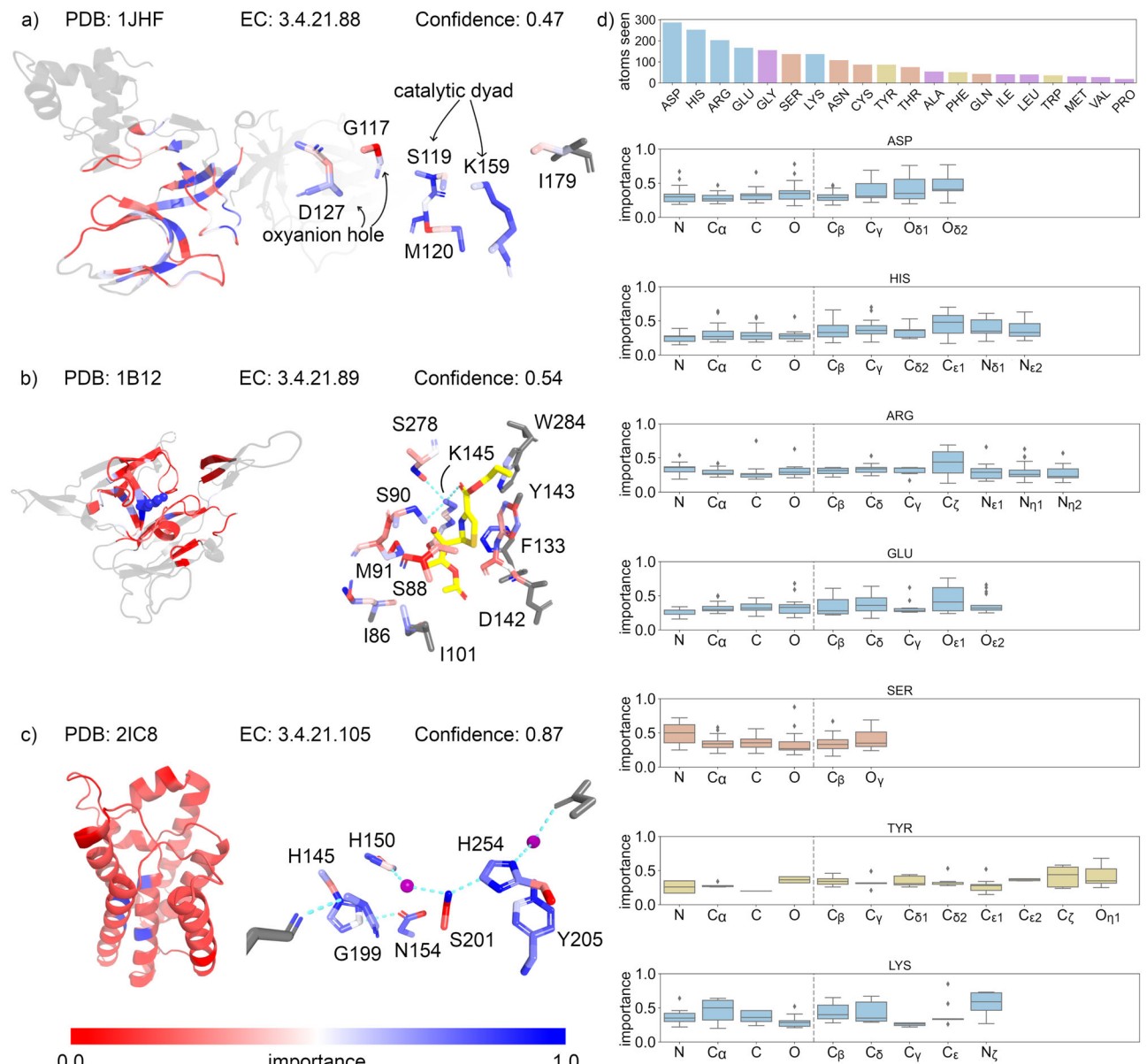

**Fig. 4 | The importance of network nodes for the prediction at atomic resolution as determined with GNNExplainer.** Each network node corresponds to the atom position within a residue. The network model is trained and tested on the same data as in Fig. 3a–g. **a**–**c** Three examples of serine endopeptidases are shown, on the left, the structure and importance as determined from the C$_\alpha$ position of the residues, on the right, the importance at atom level for residues involved in substrate binding as extracted from literature. **a** Repressor LexA, the catalytic Ser-Lys dyad (Ser119, Lys159), oxyanion hole (Asp127, Gly117) and hydrophobic packing residues (Ile179, Met120) are shown. **b** Signal peptidase I with a beta-lactam inhibitor (1PN) in complex. The network did not see the ligand. The Ser-Lys dyad (Ser90, Lys145) is shown along with the rest of the binding site residues.

**c** Rhomboid protease, the catalytic Ser-His dyad (Ser201, His254) is shown along with the rest of the binding site residues and the stabilizing waters as purple spheres. **d** The importance for all heavy atoms within the most frequent catalytic and binding residues from the CSA (Asp, His, Arg, Glu, Ser, Tyr, Lys). The bar plot shows how many atoms we have seen for every explained residue type (*n* = 142 catalytic sites). The other catalytic residues are shown in Supplementary Fig. 22. The importance of all non-catalytic and non-binding residues is shown in Supplementary Fig. 23. The box plot shows the interquartile range with the median indicated by a horizontal line. The whiskers represent the outer 25% of data points while the dots are the outliers. The confidence indicates how certain our network is of the prediction. Source data are provided as a Source Data file.

Overall, these results show that within catalytic and binding residues, the atoms most often involved in catalytic reactions are more important for prediction, indicating that the networks at atomic resolution can learn important information from specific chemical interactions. We are not able to test if this improvement in understandability is due to a more increased focus on the binding site rather than a result of including additional heavy atoms. Using random binding site locations, we will not get correct predictions to measure the explainability of the catalytic atoms. Using the full protein at atomistic resolution is not possible with the current GPUs and this model.

## Discussion

Our method presents several significant advantages and accomplishments in the realm of structure-based enzyme function prediction. First, we managed to significantly improve EC classification prediction (*F*-score: 0.72) without fold bias at residue and atomic resolutions. Importantly, our approach eliminates the need for abstraction of the 3D structure, that way preserving intricate details of the binding site region. Second, we trained networks that can classify both experimental and computationally generated enzyme structures for a vast functional space (>800 ECs). Notably, the model proves robust to

uncertainties in binding site locations, ensuring reliable performance. Lastly, the localized 3D descriptor optimizes memory consumption for 3D atomistic graphs and is suited for per-atom explainability, allowing for in-depth interrogation and revealing the significance of an atom or residue position in the prediction process. While sequence-based methods such as CLEAN can perform slightly better (F-score: 0.74), TopEC offers an alternative structural view. It has been shown that sequence-based methods can be less sensitive[71–73]. TopEC here offers an alternative tool to the enzyme function prediction toolbox, because the networks are robust with respect function prediction in the context of a chemical environment.

To achieve these results, we implemented a localized 3D descriptor to improve structure-based enzyme function prediction with GNNs. Using the localized 3D descriptor reduces fold bias and lowers the computational requirements compared to using the full enzyme structure. Still, the localized 3D descriptor becomes too memory intensive on an Nvidia A100 40 GB GPU when including more than the nearest 150–200 atoms around the binding site for training, reaching training speeds of 2.5 s per iteration using a batch size of 256. With datacenter GPUs that will contain more GPU memory and faster memory bandwidth, this shall become less of an issue. The GPU requirement is only an issue for training new models. For generating predictions with trained models, no specialized hardware is needed. We tested the predictive speed on a workstation with an Intel Core I7-10700 @ 2.90 GHz using residue-based and atomistic graphs. We can predict 100 samples with 150 nodes in only a few seconds. Full details on the performance are available in Supplementary Table 4.

The applicability of the localized 3D descriptor depends on finding the location of the binding site. With random binding site locations, the network performance drops. Notably, using contemporary binding site prediction tools, the network performance remains similar to using experimentally determined binding sites. Furthermore, we showed (Supplementary Fig. 3) that it is sufficient that the binding information is contained within the localized 3D descriptor by increasing the size of the randomly selected binding site locations. Additionally, deep neural networks as used here are generally robust to some noise in the dataset. A possible future solution could be a two-stage network: As GNNs learn to recognize local regions at residue resolution using the full enzyme object, one could use such networks as input for a higher-resolution, more localized enzyme object. This would allow us to create a network where fold information is not discarded, while it still distinguishes local molecular recognition information in the second stage.

We included AF2 structures in the TopEnzyme database to obtain more samples (>200k enzymes) over a large functional space (>5.800 EC classifications, 703 EC classifications with >50 structures). Although generally these models are of good quality, using them will introduce noise, as AF2 can overestimate the quality of its models[6] and produces lower-quality models with shallow MSA[74]. Using predicted enzyme models may lead to focusing on a single conformational state only, i.e., the one most likely represented in the PDB[3]. This can reduce the network information, as we would only see atoms for a specific molecular recognition from these static objects. By contrast, in the PDB, an enzyme can be represented in multiple conformational and binding states. On the other hand, information from the PDB might suffer from ambiguities, e.g., when an inhibitor is crystallized with an enzyme or enzyme variants in which catalytically relevant residues have been mutated are deposited albeit labeled with the EC number of the functional enzyme. Recent methods such as AlphaFlow[75] or MSA tuning[76,77] would allow for the creation of databases with multiple conformations for an enzyme based on various states of proteins. Training methods on such databases might improve the predictive quality.

Our results highlighted the importance of eliminating fold information with fold splits for the sake of generalization capabilities, even though it decreases network performance compared to when fold bias remains included. Decoupling fold information from function is also important for downstream tasks involving small local changes, such as in enzyme engineering. There are tools available for inferring enzyme function from general sequence, fold, or evolutionary information[78]. Although they are frequently successful, these methods may also propagate errors in biological databases: Recently, a study[79] showed that enzymes with EC 1.1.3.15 have been misclassified by such methods because, for certain sequences, a deviation in local structural features influences the function. Overall, this points to the need for a method such as TopEC that can discard global enzyme information and learn from local information.

Finally, we aimed to scrutinize how our networks learn information on enzyme function and what we can infer from it. We used GNNExplainer, which calculates for a GNN how important each node is for the prediction. We simulate the removal of chemical information at nodes with node masks. As in our model graph, edges are constructed based on the neighborhood radius and not on chemical bonds, generating an explanation for graph edges would not be interpretable in terms of the underlying chemical system. Thus, we disregarded edge information in our analysis.

At residue resolution, GNNExplainer reveals no differences between binding and non-binding residues related to expert knowledge based on the residues' role in function. By contrast, when classifying at atomic resolution, which contains side chain information on molecular recognition, GNNExplainer shows large differences for atoms within catalytic residues. Usually, the importance distribution of interacting atoms in catalytic residues is shifted to higher values. These results highlight the importance of including side chain positions in the graphs.

These results demonstrate how we can improve structure-based enzyme function classification with GNNs. An important contribution is the localized 3D descriptor, which allows us to reduce fold bias and reduces computational costs, allowing us to classify at higher resolution. Furthermore, the inclusion of predicted enzyme models allows us to cover a larger functional space and increases prediction performance. Finally, current explainable AI methods can be used for 3D graphs to understand how the network learns, revealing that this involves an interplay of biochemical features, such as specific interactions or conserved motifs, and local shape-dependent features.

TopEC should be a useful tool to add to the bioinformatics toolbox, especially for challenges where decoupling function from fold is important. E.g. TIM barrel structures have different functions because important residues are located on loops. TopEC could be used to investigate the importance of loop residues and make predictions based on the local chemistry. TopEC might also be a good tool for enzymes created by divergent evolution. Since these enzymes reach a similar function from different scaffolds, we should be able to classify the function if we have seen it before in different enzymes. Furthermore, TopEC can be used to screen for new enzymes in silico, e.g., by steering directed evolution methods to generate new enzymes from previous scaffolds, using TopEC to predict the change in function. Lastly, TopEC offers an alternative method for EC prediction. Our tool offers a structural view, trained specifically for cases where the enzymes might be similar in sequence and fold but different in function. This is important as many enzymes can evolve their function across enzyme hierarchies with minute changes to the structure and sequence. While the exact error rate of annotations in current databases is unknown, we have over 30 million predicted enzyme functions in the TrEMBL database. These are mainly predicted by sequence-based methods. With the advent of high throughput structure generation, TopEC can complement such predictions and possibly lower the error rate of EC annotations.

## Methods

### Data sources and enzyme classification annotation

The enzyme structures used for the networks come from various sources. Crystal structures are obtained from both the Binding Mother of All Databases (Binding MOAD, release 2020)[39] and the Protein Data Bank (PDB)[2]. From the Binding MOAD, we obtained 21,333 experimentally determined enzyme structures with an identified binding site and enzyme classifications. From the PDB, we obtained all structures submitted before 23 March 2022 where at least one chain has an enzyme classification. We do not separate the PDB structures into multiple chains, such that our localized descriptor includes information on the local chemistry from nearby chains. Furthermore, for computationally generated structures, we used the TopEnzyme database (release 2022)[6]. We developed this database to create a link between UniProt IDs, enzyme classifications, and known structures for enzymes. We removed DNA chains and only kept ATOM record data to reduce the file sizes. Crystal structures obtained from the Binding MOAD are prefixed with "BM_" before the PDB identifier. Crystal structures obtained from the PDB contain no prefix. Computational structures obtained from the AlphaFold2 DB[7] are prefixed with "AF2_" followed by the UniProt identifier. Computational structures generated by TopModel[5] are prefixed with "TopM_" followed by the UniProt identifier.

### Data partitioning

When studying main class classification, we use all obtained structures with an enzyme annotation. In the hierarchical classification experiments, we follow the procedure from DeepFRI[12]: we remove all enzyme classes with less than 50 structures to have sufficient samples for a training, validation, and test split. Training, validation, and test ratios were kept to approximately 80%/10%/10% for all splits. To test our networks in realistic use cases, we test two different data splits. First, we used a temporal data split, where the structures are partitioned by the deposition date in the PDB. In the case of the computational structures, there is no PDB deposition date. Instead, we use the deposition date of the homolog. In the case of the AlphaFold2 DB structures, we have no access to the MSA alignment or homologs used in model creation. For these structures, we do not perform a temporal data split. Second, we used a fold split. Using MMSeqs2[80], we cluster the sequence for each structure with a minimum sequence identity of 30%. Any cluster containing multiple enzyme classifications is separated by EC number. When we obtain three or more clusters per enzyme classification, we separate the clusters over the training, validation, and test sets. When we obtain two clusters, one cluster is used for the test set, while the other one is divided over the training and validation set. For enzyme classifications with only one cluster, we divide the structures such that we keep the 80%/10%/10% ratio. For each data split a.csv file is added to the repository (see "Data availability") for reproducibility. Furthermore, the repository contains a single.csv file with metadata for each enzyme used in this study.

### Binding sites

For crystal structures obtained from Binding MOAD, the binding site location based on experimental evidence is included. For crystal structures from the PDB and computationally generated structures from AlphaFold2, we find the binding sites with P2Rank. For structures generated with TopModel, we follow a different protocol, exploiting information from the homologs used in the modeling. First, we superimpose homologs with bound ligands and a TM-align[81] score >0.5 to the modeled structure. Second, we filter for common crystallization ligands and ligands not present in the binding MOAD. Third, the ligand locations in the homologs are then transferred to the modeled structure as binding site locations. The locations are ranked by the number of overlapping ligands. We take only the top 1 location as this gave the best performance. TopEC extracts the local region around the binding

site from the protein structure within the network. This allows for full customizability of the extracted region without having to edit PDB files. Currently, three methods are implemented: (1) No cutting. The full enzyme object is used in the network. This option will lead to crashes on large networks if the GPU memory is too small (<40 GB). (2) Circular cut. Every residue or atom within a configurable radius from the given binding site center is kept. (3) Count cut. The selection of atoms or residues is expanded from the closest to the furthest away from the binding site center until a defined number of atoms or residues is reached. Users can add custom extraction routines without having to make changes in the network code. An example of introducing a custom routine is given in the README file of the repository.

### Atom and residue annotation

To distinguish the chemical space among atom and residue types, respectively, we annotate each atom or residue based on the ff19SB forcefield from AMBER[82]. We furthermore extend this for heavy atoms and hydrogens into 21 annotations for residues, 31 annotations for heavy atoms, and 20 annotations for hydrogens. We planned to use the hydrogen positions for classification, however, decided against it to reduce memory consumption. Nevertheless, we kept the implementation available for users. The annotations are listed in Supplementary Table 3.

### Graph neural networks

The largest difference compared to typical Graph Convolutional Neural Networks (GCNN) is the implementation of the radial Bessel filter (RBF) in SchNet and DimeNet++ and spherical Fourier Bessel filter (SBF) in DimeNet++ (Fig. 1a), which encode the relative distances and angles between atoms in these message-passing networks. In SchNet, the atom positions are embedded in the network while RBFs are used to update the atomic representation based on the molecular geometry in the interaction blocks. The embeddings are passed through multiple interaction blocks before reaching a linear layer (also known as a fully connected layer) and a shifted Softplus activation layer before dropout and classification. In DimeNet++, the embeddings are updated in multiple interaction blocks, where each block passes the resulting embedding to an output block that transforms and sums up the output. The outputs of all embedding and interaction blocks are summed up to generate the prediction. Both models calculate probabilities for enzyme functions to be in a specific EC class represented by four hierarchical numbers.

**SchNet.** We created a SchNet implementation for proteins. Originally, SchNet was developed as a deep learning architecture for modeling quantum interactions in molecules[18]. SchNet introduced continuous filters that do not rely on discretized atom positions. Here, we briefly explain the method according to Schütt et al.[18]. Given a feature representation of $n$ objects $X^l = (x_1^l, \ldots, x_n^l)$ with $x_i^l \in R^F$ at locations $R = (r_1, \ldots, r_n)$ with $r_i \in R^D$, the continuous-filter convolutional layer $l$ requires a filter-generating function $W^l : R^D \to R^F$ that maps from a position to the corresponding filter values. The output $x_i^{l+1}$ for the convolution layer at position $r_i$ is then given by:

$$x_i^{l+1} = \sum_j x_j^l \circ W^l \left( r_i - r_j \right) \tag{1}$$

We use a similar molecular representation as described in Schütt et al. We describe $n$ atoms with specific atom types $Z = (Z_1, \ldots, Z_n)$ and atomic positions $R = (r_1, \ldots, r_n)$, which can be described as a tuple $(x_i^l = (Z_i, r_i))$ of features $X^l = (x_1^l, \ldots, x_n^l)$ with $x_i^l \in R^F$. Furthermore, Schuett et al. introduced atom-wise layers, which recombine the feature maps. These are dense layers that are applied separately to the

representation $x_i^l$ of atom $i$, with $b^l$ being the atom-wise layer:

$$x_i^{l+1} = W^l x_i^l + b^l \tag{2}$$

The filter-generating networks used in the continuous-filter convolutional layer in Schuett et al. are restricted to satisfy rotational invariance conditions. The rotational invariance is obtained by utilizing interatomic distances $d_{ij} = \| r_i - r_j \|$ expanded with a radial basis function:

$$e_k \left( r_i - r_j \right) = exp \left( -\gamma \| d_{ij} - \mu_k \|^2 \right) \tag{3}$$

located at centers $0\text{Å} \leq \mu_k \leq 30\text{Å}$ every 0.1 Å with $\gamma = 10$ Å. The distances are fed into dense layers with a softplus activation to compute the filter weights $W \left( r_i - r_j \right)$.

**DimeNet$^{++}$.** In DimeNet, equivariant directional embeddings are introduced by Gasteiger et al.[19]. DimeNet and DimeNet++ were originally developed for modeling quantum interactions in molecules similar to SchNet. Directional information associated with angles $\alpha_{(kj,ji)} = \angle m_k m_j m_i$ is leveraged when aggregating the neighboring embeddings $m_{kj}$ of $m_{ji}$. Each atom $i$ receives a set of incoming messages $m_{ji}, \sum_{j \in N_i} m_{ji}$, and updates message $m_{ji}$ based on the incoming message $m_{kj}$. The update function for the message embeddings then becomes:

$$m_{ji}^{(l+1)} = f_{update} \tag{4}$$

where $e_{RBF}^{(ji)}$ denotes the radial basis function as shown in Eq. (3). In DimeNet$^{++}$, improvements are made to the directional message passing blocks in the architecture[20]. The bilinear layer for transformation between the basis representations $e_{RBF}^{(ji)}$, $\alpha_{SBF}^{(kj,ji)}$, and the embeddings $m_{kj}$ are replaced by a Hadamard product. To compensate for the loss in expressiveness, multilayer perceptrons are introduced for the basis representations.

### Model training and hyper parameter tuning

Similarly, as in Gligorijević et al.[12], the neural networks are trained to minimize the weighted cross-entropy loss function, which gives higher weights to EC terms with fewer training examples:

$$l(x,y) = L = \{ l_1, \ldots, l_N \}^T, l_n = - \sum_{c=1}^{C} \omega_c log \frac{exp(x_{n,c})}{\sum_{i=1}^{C} exp(x_{n,i})} y_{n,c} \tag{5}$$

where $x$ is the input, $y$ is the target, $\omega$ is the weight, $C$ is the number of classes, and $N$ spans the minibatch dimension. Hyper-parameter optimization is performed using the Optuna sweeper[83] available within the Hydra framework[84]. The Hydra framework allows the creation of simple configuration files for experiments with complex settings. The search space for Optuna is configured to perform a grid search within the parameter space. To avoid overfitting, we use an early stopping criterion with a validation/accuracy patience of 10. That is, we stop training the networks if the accuracy on the validation set does not increase after 10 epochs. As hyperparameter trends tend to generalize to new datasets[85,86], we perform the optimization on the networks by combining the Binding MOAD and TopEnzyme data to reduce the time spent on hyperparameter optimization. This dataset covers both experimental and computational structures with accurate binding sites. We use the Adam optimizer with $lr = 0.001$, $\beta_1 = 0.9$ and $\beta_2 = 0.999$. We randomly translate atom positions by 0.05 Å in the training set to improve the network performance. The dropout is set to 0.25. In the SchNet implementation, we use a six-layer network with 128 graph embeddings and filters. The hidden embedding size is set to 128. We use 50 Gaussians with 32 maximum neighbors per node. In the DimeNet++ implementation, we use 128 graph embeddings with a hidden embedding size set to 128. A total of seven spherical harmonics

and six radial basis functions are used to build the Fourier-Bessel filters as described in Gasteiger et al. The network consists of four building blocks (embedding + interaction blocks) where the basis embedding is eight and the interaction block embedding is 64. Within the interaction block, the size of the output embedding is 256. The batch size varies per method, from the smallest to the largest localized 3D descriptor we can increase the graph size hundred-fold. We use the highest batch size in a geometric sequence with a common ratio of two fitting alongside the model on an Nvidia A100 40 GB GPU. The entire method has been implemented using the PyTorch[87], PyTorch Lightning[88], and PyTorch geometric[89] libraries.

**GNNExplainer.** As described in Ying et al.[56], GNNExplainer generates an explanation of important nodes for the correct classification by identifying the most influential subgraph and subset of node features in the model prediction. As we already identify the subgraph structure best describing the enzyme function (binding pocket), we forgo this step in GNNExplainer. To generate an explanation, GNNExplainer learns an edge mask M and a feature mask F by optimizing the following objective function:

$$I(y, \hat{y}) + \alpha_1 ||M|| + \alpha_2 H(M) + \beta_1 \tag{6}$$

where $l$ is the loss function, $y$ is the original model prediction, $\hat{y}$ is the model prediction with the edge and feature mask applied. $H$ is defined as the entropy function (a detailed explanation is available in Ying et al.). $\alpha_1$, $\alpha_2$, $\beta_1$, $\beta_2$ are tunable parameters. Higher $\alpha_1$ and $\alpha_2$ values will make the explanation edge masks more sparse by decreasing the sum of the edge mask and the entropy of the edge mask, respectively. Higher $\beta_1$ and $\beta_2$ values will make the explanation node feature masks more sparse by decreasing the mean of the node feature mask and the entropy of the node feature mask, respectively. We do not encode edge masks as our graph edges do not represent chemical bonds in the system. Instead, we optimize the following objective function:

$$I(y, \hat{y}) + \beta_1 \tag{7}$$

where the parameters are similar as in Eq. (6) with the entropy function:

$$H = - F log(F + c) - (1 - F) log(1 - F + c) \tag{8}$$

where $c$ is a small positive constant to avoid a log of 0.

### Reporting summary

Further information on research design is available in the Nature Portfolio Reporting Summary linked to this article.

## Data availability

The raw PDB files, a hierarchical data format (HDF5) of the structures, source data for this paper, and the trained networks used in this study are available in researchdata.hhu.de under accession code https://doi.org/10.25838/d5p-66[90]. Source Data are available with this paper as a Source Data file.

## Code availability

All code is available at https://github.com/IBG4-CBCLab/TopEC under a CC BY-NC-SA 4.0 license. The code is also deposited in researchda-ta.hhu.de at https://doi.org/10.25838/d5p-66[90] for archiving purposes, to make the code citable and to improve reproducibility.

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

## Acknowledgements

This work was supported as part of the Helmholtz School for Data Science in Life, Earth, and Energy (HDS-LEE) (Helmholtz Association of German Research Centers, funding number HIDSS-0004, H.G.). We gratefully acknowledge the computing time provided by the John von Neumann Institute for Computing (NIC) on the supercomputers JUWELS and JUR-ECA at Jülich Supercomputing Centre (JSC) (user IDs: VSK33, found).

## Author contributions

K.v.d.W.: conceptualization, data generation, software, calculations, analysis, visualization, manuscript preparation, project management; E.M.: conceptualization, software, analysis, manuscript review; M.P.: supervision, manuscript review, funding, resources; H.G.: conceptualization, analysis, manuscript preparation, supervision, funding, resources.

## Funding

## Competing interests

The authors declare no competing interests.
