## [Transparent Peer Review file · Nature Communications]

TopEC: Prediction of Enzyme Commission classes by 3D Graph Neural Networks and localized 3D protein descriptor

Corresponding Author: Professor Holger Gohlke

Version 0:

Reviewer comments:

Reviewer #1

(Remarks to the Author)
see attached

(Remarks on code availability)

Reviewer #2

(Remarks to the Author)
The paper by van der Weg et al. addresses a timely subject of enzyme function classification, which recently attracted attention in the literature, likely due to the importance of this problem, progress in computations, and machine learning. The authors present some interesting results and conclusions, albeit in a rather scattershot paper.

major comments

=====
(comments ordered according to importance)

The most obvious, glaring omission is the lack of comparison to recent state-of-the-art methods for enzyme function prediction - PARSE (<https://www.ncbi.nlm.nih.gov/pmc/articles/PMC10614799/>) and CLEAN (<https://www.science.org/doi/10.1126/science.adf2465>). While I can accept that PARSE is still a pre-print, therefore I would not expect to see a comparison (though, I would welcome it), CLEAN was published almost a year ago (30 Mar 2023). I would expect those methods to be included in the benchmarking or, at the very least discussion of why, in the opinion of the authors, such a benchmark is not justified. Otherwise, it is hard to relate TopEC to state-of-the-art.

Technical requirements of the method are not discussed transparently. Throughout the manuscript, the authors hint at the fact that the all-atom representation for the model does not scale well (e.g. lines 92-93, 133-134, 137-140, 253-254, 459-460, 568-572). However, considering this I would expect a clearer disambiguation and discussion of those computational requirements.

* Should the users always just use the Calpha representation or are there cases where an all-atom representation should be used? How computationally intense is the method using a Calpha representation?

* I understand that computational power increases rapidly, so a computation which takes hours today may take minutes a year from now. Even a head-to-head comparison between methods would suffice.

* Providing runtime (ideally, as a function of protein size) would help the end-user estimate how the proposed models scale in time.

* Also, the fact that the proposed models may be run on GPU only, reduces their usability. Authors are welcome to discuss in more detail how much computational power is required depending on models i.e. SchNet, DimeNet++; types i.e. residue, atom; and different thresholds.

There are 7 datasets used in this study: BindingMOAD, Combined, PDB300, AF703, Price, CSA and PrOSPECCT. It's unclear why some datasets are used in some analyses but not others. A table summarizing the datasets (ok with a supplementary table) would help and also some explanations why some datasets are used only in benchmarking, while others in analyses only. Considering the claim of relative paucity of enzyme function data, it is difficult to follow without justification. Also, it raises suspicions that TopEC may perform well on some datasets (shown as benchmark), while underperforming on other datasets.

What is TopEC, really? It's a relatively easy issue to address but a major oversight, in my opinion. The first mention of TopEC is in the title, abstract and introduction. Then, for 10 pages of the manuscript the name is gone till page 14. Instead, SchNet and DimeNet++ are mentioned prominently in the beginning of the Results. Clearly, TopEC uses SchNet or(?)and(?) DimeNet++ networks as a part of the algorithm, but it's never clearly explained. Please, be more upfront about it.

What would TopEC produce if we give it non-enzymes? Do the users need to know a priori if the query protein is an enzyme?

336-337 "Generally, we predict an enzyme class either completely correct or wrong, leading to the hourglass-shaped AUPR curves." this is a major observation that lacks discussion it deserves.

457-462 Could the authors explain in more detail why they decided to analyze explainability based on a much less-performing model (apart from technical issues)? To shed more light on the way how the model works (and hence reduce the methodological bias), it would be fair to analyze also some negative cases (i.e. when models gave incorrect predictions).

minor comments

=====

(comments made in the manuscript order)

31 "molecular function TO enzymes" should be "molecular function OF enzymes".

40-41 "significantly improved EC classification" - the sentence doesn't say with regards to what the classification is improved.

63-67 How does the statement "often molecular function cannot be deduced directly from the structural representation, or the enzyme sequence is annotated incorrectly in databases" relate to the subsequent sentence i.e. how computational methods overcome these experimental limitations?

68-70 Among examples of the use of GNN function prediction seems to be missing. deepFRI (ref. 25) is one such example, against which TopEC is even benchmarked.

97 "F1-score of 0.71"... in the abstract (line 41) the reported value is 0.72. It's also unclear where this number comes from. Table 1 never reports a value of 0.71 or 0.72

Fig. 1a abbreviations RBF and SBF are not explained in the figure caption. They're only explained towards the end of the manuscript in lines 749-751.

Fig. 1a overlaying DimeNet++ and SchNet architectures is space-efficient but I find it hard to follow what exactly are the differences between them. Presenting the architectures on top of each other could improve legibility.

112-113 "Residues colored in blue are selected for the input, whereas those in red are discarded." this sentence is a part of the caption for (b) but I believe it refers to (c) and (d). Otherwise, it makes little sense to me why Ser would be considered and Thr discarded.

119 What is "fold bias"? the term is used in the abstract and here for the first time. It seems that the authors have a clear intention.

132 The way point (1) is phrased is difficult to follow. The authors intend this to be a positive thing but sounds negative.

147 Data S1 and Data S2 are on GitHub but their location is not mentioned in the text.

161 and other places - BindingMOAD is inconsistently named. Sometimes it's referred to as Binding MOAD and sometimes BindingMOAD.

164 add "For the fold split" we used MMSeqs2.

164 "cluster out database BY 30%" should be "cluster our database AT 30%".

164 The term "fold split" used throughout this manuscript is somewhat misleading. First, because the authors in fact perform

a sequence-based clustering at 30% sequence identity, which is a sequence-based and not a “fold” split. Since for all evaluated datasets 3D structures exist (either experimental or models) a real fold split should rely on structures not sequences. Second, the datasets used are on the orders of 10,000s entries - MMSeqs2 is fast but not accurate at low sequence identity values, so using a more exact method (e.g. CD-HIT) seems to be feasible on those datasets.

187-188 “Structure-based predictions...” this sentence seems out of place here. Please, either clarify or remove.

189 + also Fig. 2D what are the DS1-DS7 categories? Later on it's alluded to (e.g. lines 338-339, 349) but it's never explained in detail.

Table 1C why only 2 decimal places while in all other sections of Table 1 there are 3 decimal places reported?

220-221 Please, substantiate the claim by providing a suitable reference. Also, the sentence seems to stand in contradiction to eg. 132-133, 241-243, 257-258.

221-223 Sizeably better performance for temporal split as compared to fold split in the case of TopEC models holds only for BindingMOAD and combined datasets but is less evident for the TopEnzyme dataset. Why? Moreover, such discrepancy is not observed for DeepFRI. Does it mean that DimeNet++ and SchNet are more dependent on structural homology than other methods, eg. DeepFRI?

Fig. 2 Labels (a)-(d) are a part of the title of each plot and thus are poorly legible.

413 Would it be possible to relate residue translation to experimental resolution?

Fig. 3g presented 3D structures are too small to appreciate the insights that the authors arrive at.

458-460 It's unclear if the drop in the number of correct predictions stems only from the fact that the method requires more computing or are there any other factors at play? Did TopEC just fail to compute predictions in 90 cases where all-atom is worse? Or do the authors just speculate that it's worse because we're not able to use as many atoms in the descriptor? One possible experiment would be to limit the Calpha descriptor to the same atoms which the all-atom model could compute, then compare. I would like to see some evidence for one or the other.

464-527 Paragraphs provide an eloquent description of what may be observed from TopEC results but are those insights correct? I am lacking some (more) references to the literature that would substantiate the observations made on the basis of TopEC predictions.

557-558 The statement may be true (compare comments to lines 458-460) but also the authors repeatedly claim that the method does not scale.

577-579 I read through the whole paper at this point but I am still not sure if TopEC does it automatically out-of-the-box or are the users deferred to 3rd-party tools to predict binding sites?

592-593 As far as I am concerned, all databases, save for NMR data, have a single conformational state for any given entry, so how is that a flaw of AF2? Also, there is some work showing how to use AF2 to predict multiple conformational states (e.g. <https://www.nature.com/articles/s41586-023-06832-9> or <https://www.sciencedirect.com/science/article/pii/S0959440X23001197>). Please, clarify.

614-615 The authors say that GNNExplainer doesn't really work since it's designed for 2D GNNs and then go on in lines 616-633 to discuss its results. I find a lengthy discussion of a 3rd party tool which according to the authors “might not generalize as well to 3D GNNs” out of place here.

(Remarks on code availability)

The link in the section above is incorrect but indeed I looked at the code. I have not tested it on real data, though. Overall, the repo is well-structured and organized. There are no unit tests but the code is well laid-out and each function or class is documented with a docstring.

Reviewer #3

(Remarks to the Author)

(Remarks on code availability)

The repository contains the code itself, supplementary information, training datasets, notebooks, and README containing details about the software e.g. basic information, dependencies, and how to run the code. Although I haven't installed the code and run it by myself, it seems that, at least partially, the results in the paper may be reproduced based on the code.

Version 1:

Reviewer comments:

Reviewer #1

(Remarks to the Author)

The authors have addressed my concerns

(Remarks on code availability)

Reviewer #2

(Remarks to the Author)

The authors have made significant efforts to enhance the appeal and clarity of their manuscript. After revisions, the work appears more comprehensive, and the method itself is now more transparent. I appreciate the comparison to the CLEAN method, which demonstrates that both approaches offer comparable performance despite utilizing different inputs. While it's unfortunate that the authors were unable to run PARSE, given that this is a pre-print, its omission is acceptable. However, for the future, I strongly encourage reaching out to the authors for assistance in case of problems, as I believe thorough comparisons work in favor of paper's longevity and relevance. I acknowledge the addition of runtime benchmarks (Table S4) and the clearer dataset description (Table 1). All minor and almost all major comments appear to have been addressed satisfactorily.

I would like further clarification on the following points:

[258-265] I encourage the authors to conduct a comparative analysis to illustrate how 30% sequence identity translates into Foldseek's structural clustering. Ideally, a small test could be added to show how TopEC performs on folds derived using Foldseek. Despite Authors' clarification that the tool was not available at the time of work, Foldseek has been available as a full Nature Biotechnology paper for over a year now (10 months at the time of 1st review) and the tool has been widely used by the community for 2.5 years already (since Feb 2022).

[513-519] Providing an explainability analysis for the Ca descriptor that leads to more accurate predictions as compared to an all-atom based approach would give valuable insights into how the method operates — specifically, which backbone atoms are utilized in the inference step. Additionally, the authors could discuss whether the interacting atoms identified by this analysis hold biological significance.

[595-598] Based on the statistics and conclusions drawn in this section, is it feasible to add a refinement step by assigning greater importance to certain interacting atoms, potentially improving the accuracy of functional predictions?

(Remarks on code availability)

Reviewer #3

(Remarks to the Author)

(Remarks on code availability)

The URL mentioned above is incorrect, but the authors have provided the correct URL within the manuscript: <https://github.com/IBG4-CBCLab/TopEC>. They have updated the repository based on the reviewers' suggestions. However, I cannot confirm whether the code functions properly, as I have not personally run it.

Version 2:

Reviewer comments:

Reviewer #2

(Remarks to the Author)

I thank and applaud the authors for addressing all of the concerns well!

(Remarks on code availability)

The repository is generally well-structured, exhaustive and contains an useful README.
Some minor remarks about the GitHub repo:

1. following the installation instructions provided in the README file does not work without fixing. The `requirements.txt` file

seems to be not specified correctly (some conflicts exist) and some dependencies also seem to be missing (e.g. Biopython which is required by some of the modules). I was able to make it work with some minor tinkering but could be easily fixed by testing on a clean environment and/or including channels in the specification + not over-specifying some software versions (e.g. I wasn't able to install with `torchvision==0.16.2+cu121` which assumes CUDA 11.2)

2. some Python scripts in the main folder are not described in the readme, e.g. `prepare_data.py` or `run_dataset_create.py`

Reviewer #3

(Remarks to the Author)

(Remarks on code availability)

The code appears complete and professionally organized, with clear and detailed instructions for retraining the model and making inferences. However, I encountered dependency conflicts while attempting to run the code during the creation of the virtual or Conda environment. I recommend that the authors enhance the reproducibility of the Python environment. This could involve upgrading the Python version from 3.9 to a more recent one or providing a YAML file as an alternative to the requirements.txt file.

Responses to the reviewers' comments

We are very grateful to the reviewers for pointing out important points, which we have addressed as detailed below. We have prepended each comment and answer with **REVIEWER #** and **TopEC** to mark comments and responses.

REVIEWER 1:

In this paper, van der Weg et al. present TopEC, a graph neural network for predicting EnzymeCommission classes based on three-dimensional enzyme structures. The proposed network is based on modern SchNet and DimeNet++ architectures, allowing it to learn from complex relationships between biochemical features and shape-related characteristics of enzyme structures. The resulting models show impressive performance under certain circumstances and present exciting opportunities for improving structure-based function classification of enzymes. Our chief concerns are over the apparent lack of novelty in the proposed method, as well as the paper's difficulty in showing its improvement over the state of the art. Our comments and suggestions are described in greater detail below.

TopEC:

Thank you for your feedback. We addressed your concerns over the lack of novelty by more explicitly detailing the novel components of our localized descriptor and the framework of TopEC for designing new experiments.

Furthermore, to improve the comparison with the state-of-the-art, we trained CLEAN on the largest dataset used in our study for comparison. Interestingly the performance of CLEAN (F1-score: 0.74) is only slightly higher for unseen enzymes compared to TopEC (F1-score: 0.72). CLEAN is a sequence-based method built on top of ESM2-1b, which contains pre-trained knowledge of more than 250 million sequences. In comparison, our network is only trained on 250 thousand protein structures. We added more discussion on the different biases of sequence- and structure-based methods and view TopEC as a complementary method to sequence-based methods (please see below for details).

REVIEWER 1:

1. Please elaborate more on the contribution of the paper. A significant component appears to be the combination of pre-existing models; however, that is of limited interest. If the authors made modifications to the models or introduced new techniques of their own, a more explicit description would help establish the novelty of their work.

TopEC:

Thank you for your comment. In our view, a novel component is the use of a localized 3D descriptor at different resolutions as input to the network, allowing us to investigate how the networks can learn. On top of this, we adapted atom typing from Molecular Dynamics forcefields, e.g., ff19SB, to distinguish similar elements in a variety of local chemical environments. This last point is only briefly mentioned in the Methods section; to provide greater emphasis, we added the following sentence to the Introduction at lines 100-102:

“We encode the atoms and residues following the atom type definitions in the force field ff19SB, allowing the network to learn different local chemical environments for one element.”

Furthermore, we realized that ML software can be hard to grasp from an end-user point. We made a framework in which users can create and make their own experiments for enzyme function prediction by changing parameters in the experiment file, without having to have knowledge of the ML models. This framework also allows end users to (re)train new models with new sets of functions. Slightly more experienced users can easily add in new graph models, without having to (re)write the code for most of the project, such as the data loading, pre- and postprocessing, training loops, metric collections, and metric reports. An example of this is shown in the repository for graph interaction networks, showing how we can test new graph networks with as little as 40 lines of code. This flexibility is important to experiment with, refine, and better understand what deep learning models can do from a biochemical viewpoint. Thus, besides the training and combination of pre-existing models, TopEC is a full experimentation framework made to be more accessible to bioinformaticians with less experience in deep learning. We added more information on how to accomplish this on the GitHub repository. We stressed this in the text by adding the following sentences at the end of the introduction (lines111-116):

On top of this, TopEC is built as a framework for rapidly devising, training, and testing new deep-learning experiments based on protein structures. The software is written such that users can control the experiment from simple parameter files. Users can use TopEC to create their own function prediction tools or test different graph models with our pipeline for enzyme function prediction. TopEC is available as a repository, including accompanying data, on GitHub: <https://github.com/IBG4-CBCLab/TopEC>.

REVIEWER 1:

2. Please show more clearly how the model outperforms the current models (EnzyNet and DeepFRI) and how it compares for the larger datasets (PDB). Specifically, please include an additional performance comparison with EnzyNet on the PDB dataset, which was reported to be 78.4% in previous literature.

TopEC:

We investigated the EnzyNet dataset again and chose not to retrain DeepFRI and TopEC on this dataset. EnzyNet shows impressive performance, however, the split is designed by random allocation of structures to the training, validation, and test sets. This allows for data leakage between similar structures in the PDB, which boosts the performance on data points within the training distribution but lowers the performance on data points outside the training distribution. In comparison, we previously tried a random split on our largest dataset, and this led to a 97% prediction accuracy on the test set for the main class of EC numbers. We did not include this result in the manuscript as it is not indicative of a realistic performance.

To accommodate your request, we instead retrained EnzyNet on the most comprehensive dataset of AlphaFold and PDB structures (257,165 structures covering 826 EC designations) split by 30% sequence identity. This gives us an F1-score of 0.50 for EnzyNet compared to 0.72 for TopEC.

We tried to train DeepFRI on this dataset, but the training method breaks down for datasets with a large number of classes. Since DeepFRI optimizes for positive and negative probabilities of EC/GO terms, a large amount of negative classes pushes the network to predict a negative probability for each EC term. We tried various changes including sampling (weighted, over-/ undersampling), changes in loss (CategoricalCrossentropy, CategoricalFocalCrossEntropy, Hinge) and optimizers (Adam, AdamW, SGD), network configuration (varied depth in GConv, Dense and Dropout layers), networks (GAT/GraphConv/MultiGraphConv/SAGEConv/ChebConv/CNNs), as well as different contact map types ($C\alpha$, $C\beta$) and cutoffs (5, 10, 15, 20). None of these combinations worked to train DeepFRI successfully as the network still tends to learn the negative probability for each incorrect EC term over the positive probability for each correct EC term.

Furthermore, we also compared TopEC to CLEAN, the current state-of-the-art for EC function prediction, which reached an F1-score of 0.74 on this dataset. We added the results for this in the manuscript at lines 352-356:

We reached an F1-score of 0.72 for the combined dataset at the designation level. We tested this dataset on EnzyNet and CLEAN⁵⁴ and obtained F1-scores of 0.50 and 0.74, respectively. We were not able to obtain satisfying results for DeepFRI as this method optimizes for both positive and negative probabilities, which tends to overfit on the negative probabilities in the case of a large number of classes.

REVIEWER 1:

3. The authors note that few methods use explicitly encoded 3D information for predicting EC numbers. What are the reasons for such scarcity? Elaborating more on this point will help establish the importance of the problem being addressed.

TOPEC:

There are two reasons for this scarcity. The first one is straight-forward: Only since AlphaFold2, we have enough structural protein data to develop deep learning applications for enzyme function prediction at all four levels of classification. As the data space is complex, we need many examples to train/learn from. While the PDB has many protein structures, of the current 220,113 available, only 100,430 have an accompanying EC number. From these enzyme structures, many are redundant and cover a small subset of enzyme functions. AlphaFold2 and the AlphaFold2DB gave access to generally good enzyme structure models and allowed us to significantly expand the function space from a structural point of view.

The second reason for this scarcity is the hardware requirements for the experiments. Encoding 3D information requires a lot of GPU memory. Current available GPUs that can fit 3D atomistic details on the GPU are Nvidia A100 40Gb or more recent cards. Even then, depending on the graph architecture these cards still lack GPU memory to encode large proteins in full atomistic details.

However, once these networks are trained, users can use the trained networks to make predictions within seconds on a GPU or CPU.

We elaborated on the scarcity and rewrote the section at lines 94-97.

In EnzyNet²⁶, an abstraction of the protein to the 3D backbone information is used, and in DeepFRI²⁵, a graph is constructed from the 2D contact map created from the protein structure to reduce memory requirements compared to explicitly encoding 3D information. Furthermore, modern structure prediction methods, resulting, e.g., in the creation of the AF2 database⁷, provided enough training samples to study structural enzyme function prediction at all four levels of classification.

REVIEWER 1:

4. Performance on the Price dataset seems to leave more questions than answers. This is indeed a difficult test case, but to what extent and in what ways? How does the proposed model compare to other methods?

TOPEC:

We came across this benchmark while reading the publication on ProtelInfer¹, a sequence-based tool for enzymatic function prediction. The Price dataset comes from a study on mutant phenotypes for bacterial genes of unknown function². Interestingly, they discovered that for certain groups the annotations in the Swiss-Prot were incomplete or wrong compared to their experimental results². Even though Swiss-Prot is curated, it contains many proteins annotated only based on *in silico* methods, which turned out to be incorrect for these cases. For many enzymes, where the associated sequence is conserved, the function is similar if the sequence identity is high, however, this is not always the case. The authors of Price remark that many proteins in this dataset are not conserved². On top of this, in ref. [3], we see how minor deviations in sequence can make relatively minor deviations in structure, but large deviations with respect to function.

ProtelInfer also performs poorly on this dataset with only 12% correct predictions. Yet, a direct comparison might not be valid, as we are not able to train on all enzyme classes in the Price dataset. When there is only a single fold known for an enzyme class, we would have data leakage if we split this fold over training and testing sets.

We know that correct predictions on the Price dataset are difficult to obtain with current *in silico* tools, but a larger investigation into wrongly predicted proteins would be needed to better understand why.

We added more details on the Price dataset at lines 370-374.

Enzymes of the Price dataset are less conserved, and many of the enzymes have been previously misclassified by *in silico* methods³⁹. Similarly to ProtelInfer, we try to recover the new corrected classifications. Compared to ProtelInfer, we predict more enzymes correctly, but also more enzymes wrongly, as ProtelInfer contains a mechanism to not make uncertain predictions.

¹<https://google-research.github.io/proteininfer/>

²<https://www.nature.com/articles/s41586-018-0124-0#citeas>

³<https://journals.plos.org/ploscompbiol/article?id=10.1371/journal.pcbi.1009446>

REVIEWER 1:

5. Regarding the ProSPECCTs dataset, the contrast between poor accuracy on DS1 and good accuracy on DS5 and DS6 begs more explanation. Both pertain to structures with significant structural and ligand similarities, yet the performance is very different. What may be the reasons for this difference?

TOPEC:

Upon further investigation, we found that the performance of DS1 and DS1.2 is good. All samples for three of the four enzyme classes present in these datasets were correctly predicted. However, most of the test samples belong to the fourth enzyme class (EC: 3.6.4.10), which is wrongly predicted, skewing the results. Upon investigation, we found that 9 of the 13 unique designations in sub-subclass 3.6.4 have been moved to the Isomerase main class. These incorrect training labels propagated through the database and might lead to lower performance on this fourth enzyme class in DS1 and DS1.2. Thus, the ambiguity of information and unequally distributed enzyme classes led to poor performance.

We changed lines 380-384, which now read:

Although the overall performance shown in Figure 2d is only fair, the performance on structures with identical sequences (DS1) and similar ligands (DS1.2) should be more akin to DS5 and DS6. Only for one structural group, the wrong EC was predicted. This group contained significantly more samples, skewing the results.

REVIEWER 1:

6. The improvement in understandability seen with the finer-grain atom resolution approach may have been due to the increased focus on the binding site and not necessarily a result of the inclusion of additional heavy atoms. Can you comment on that possibility?

TOPEC:

This is possible; however, we cannot test this currently as one would need to compare to the full protein at atomistic resolution, which is not attainable for 3D GNNs with current GPUs. We added a comment on this in the manuscript.

We are not able to test if this improvement in understandability is due to a more increased focus on the binding site rather than a result of including additional heavy atoms. Using random binding site locations, we will not get correct predictions to measure the explainability of the catalytic atoms. Using the full protein at atomistic resolution is not possible with the current GPUs and this model.

REVIEWER 1:

7. While there are many potential uses for this method, whether it'll perform well in such diverse applications remains an open question. Please comment why you think TopEC is likely to be an effective approach to the proposed applications, based on the observed results or the intrinsic design of the model.

TOPEC:

We have seen experimental studies describing that small changes in structure lead to functional changes (e.g., in actin isoforms¹, chromatin variants², and (S)-2-hydroxyacid oxidases³), which can be hard to grasp from sequence information. In our method, we explicitly encode relative 3D positions between residue or atom nodes in

the network. While a small change to a sequence might have a less pronounced effect on the latent representation, a change to the structure should have a more pronounced effect on the network embeddings. We see that in our model, in cases where the local embedding changes, such as the PrOSPECCT benchmarks, the correct function can still be recovered and that the networks are robust with respect to function prediction in the context of a chemical environment. We added the following to the discussion in lines 620-625:

While sequence-based methods such as CLEAN⁵² can be slightly better (F-score: 0.74), TopEC offers an alternative structural view. It has been shown that sequence-based methods can be less sensitive⁷⁰⁻⁷². TopEC here offers an alternative tool to the enzyme function prediction toolbox because the networks are robust with respect to function prediction in the context of a chemical environment.

¹ <https://www.nature.com/articles/s41467-023-43653-w>

² <https://www.sciencedirect.com/science/article/pii/S2001037022005591>

³ <https://journals.plos.org/ploscompbiol/article?id=10.1371/journal.pcbi.1009446>

REVIEWER 1:

Some minor concerns

8. Line 82-345: six instances of “subsub-class” or “subsubclass” should be “sub-subclass.”

9. Line 303-345: four instances of “mainclass” should be “main class.”

10. Line 776: “atom l” should be “atom i.”

TOPEC:

Thank you for spotting these minor concerns. We corrected all occurrences in the manuscript.

REVIEWER 2:

The most obvious, glaring omission is the lack of comparison to recent state-of-the-art methods for enzyme function prediction - PARSE (<https://www.ncbi.nlm.nih.gov/pmc/articles/PMC10614799/>) and CLEAN (<https://www.science.org/doi/10.1126/science.adf2465>). While I can accept that PARSE is still a pre-print, therefore I would not expect to see a comparison (though, I would welcome it), CLEAN was published almost a year ago (30 Mar 2023). I would expect those methods to be included in the benchmarking or, at the very least, discussion of why, in the opinion of the authors, such a benchmark is not justified. Otherwise, it is hard to relate TopEC to state-of-the-art.

TOPEC:

Originally, we wanted to offer the structural approach for the prediction of EC function as a complementary method to sequence-based prediction tools. In our view, there is room for both methods. Sequence-based methods can be trained on much larger data spaces such as the TrEMBL and MGnify datasets. However, sequence-based methods might be less sensitive¹ as small changes in sequence can lead to large changes in function². In structure-based methods we often lack training samples, for example, the PDB is 3-4 orders of magnitude smaller than TrEMBL and MGnify.

However, with the local 3D descriptors, we can encode the chemistry of enzyme function in more fine-grained detail.

We do agree that it is hard to relate TopEC to the state-of-the-art by excluding CLEAN. We retrained CLEAN on the largest dataset we used (AF703 and PDB300 combined) and found an F1-score of 0.74 compared to 0.72 for TopEC. Unfortunately, we were not able to make PARSE working. The method for recreating the training database is not working as downloaded from <https://github.com/awfderry/PARSE>, and the required files for testing PARSE led to a “file not found” error on the Zenodo repository.

We added the CLEAN results in line 352-356:

We reached an F1-score of 0.72 for the combined dataset at the designation level. We tested this dataset on EnzyNet and CLEAN⁵² and obtained F1-scores of 0.50 and 0.74, respectively. We were not able to obtain satisfying results for DeepFRI as this method optimizes for both positive and negative probabilities, which tends to overfit on the negative probabilities in the case of a large number of classes.

Furthermore, we discuss the differences between CLEAN and TopEC in more detail in lines 604-607:

While sequence-based methods such as CLEAN⁵² can be slightly better (F-score: 0.74), TopEC offers an alternative structural view. It has been shown that sequence-based methods can be less sensitive⁷⁰⁻⁷². TopEC here offers an alternative tool to the enzyme function prediction toolbox.

¹<https://academic.oup.com/bioinformatics/article/35/20/3970/5426703>

²<https://journals.plos.org/ploscompbiol/article?id=10.1371/journal.pcbi.1009446>

REVIEWER 2:

Technical requirements of the method are not discussed transparently. Throughout the manuscript, the authors hint at the fact that the all-atom representation for the model does not scale well (e.g. lines 92-93, 133-134, 137-140, 253-254, 459-460, 568-572). However, considering this I would expect a clearer disambiguation and discussion of those computational requirements.

* Should the users always just use the Calpha representation or are there cases where an all-atom representation should be used? How computationally intense is the method using a Calpha representation?

* I understand that computational power increases rapidly, so a computation which takes hours today may take minutes a year from now. Even a head-to-head comparison between methods would suffice.

* Providing runtime (ideally, as a function of protein size) would help the end-user estimate how the proposed models scale in time.

* Also, the fact that the proposed models may be run on GPU only, reduces their usability. Authors are welcome to discuss in more detail how much computational power is required depending on models i.e. SchNet, DimeNet++; types i.e. residue, atom; and different thresholds.

TOPEC:

Thank you for your comment, we acknowledge that the technical requirements have been confusing in the discussion. The technical requirements are quite significant for training, but users should not be worried about the computational cost for making predictions. We tested the prediction speed using only the CPU in our workstation (Intel Core i7-10700 @ 2.90GHz) and found that atomistic graphs are about 1.5-2.5 times as slow to predict as C α -based graphs. For atomistic graphs, we can predict 100 samples in 5.85 s \pm 237 ms for 150 nodes (mean \pm std. dev. of 7 runs, 1 loop each). For residue-based graphs, we can predict 100 samples in 2.63 s \pm 114 for 150 nodes. We added the benchmarks to the SI in Table S4 along with the average training time per epoch for atomistic and C α -based graphs. We do recommend the user to use the C α representation to make predictions if they have no prior knowledge of the enzyme, as this model performs better. We recommend the atomistic graph if you want to use the explainability method to gain insights into the possible mechanism of enzymatic activity. We rewrote the discussion to focus more on the technical requirements for using the trained models instead of training the models. The full section in the discussion now reads (lines 626-634):

Still, the localized 3D descriptor becomes too memory intensive on a Nvidia A100 40 GB GPU when including more than the nearest 150-200 atoms around the binding site for training, reaching training speeds of 2.5 s per iteration using a batch size of 256. With datacenter GPUs that will contain more GPU memory and faster memory bandwidth, this shall become less of an issue. The GPU requirement is only an issue for training new models. For generating predictions with trained models, no specialized hardware is needed. We tested the predictive speed on a workstation with an Intel Core i7-10700 @ 2.90GHz using residue-based and atomistic graphs. We can predict 100 samples with 150 nodes in only a few seconds. Full details on the performance are available in Table S4.

We removed the following section from the discussion:

Furthermore, Fully Sharded Data Parallel (FSDP)⁶² now offers parallelism with shared memory across GPUs. Alternatively, we could have implemented abstractions to reduce compute requirements; however, not doing so offers the benefit that our network is an explainable AI model, which we can interrogate to reveal the importance of atom or residue positions.

REVIEWER 2:

There are 7 datasets used in this study: BindingMOAD, Combined, PDB300, AF703, Price, CSA, and ProSPECCT. It's unclear why some datasets are used in some analyses but not others. A table summarizing the datasets (ok with a supplementary table) would help and also some explanations why some datasets are used only in benchmarking, while others in analyses only. Considering the claim of relative paucity of enzyme function data, it is difficult to follow without justification. Also, it raises suspicions that TopEC may perform well on some datasets (shown as benchmark), while under-performing on other datasets.

TopEC:

We certainly did not mean to raise suspicion about TopEC performance being good on some datasets and worse on others. We agree this can be made clearer and added a table to the manuscript dataset section.

We added Table 1 to the manuscript, explaining the choice of datasets for BindingMoad, Combined, PDB300, AF703, and benchmark sets Price, CSA, and PrOSPECCT.

The datasets are a consequence of the experiments we performed. Originally, we used BindingMOAD because of its experimental validation of binding sites. Then we started generating homology-based models using our in-house method TopModel. For this method, we know the homology cut-offs where we can expect good-quality models, and we can extract the binding information from the homologs. We combined this together to get more data. We originally benchmarked on these datasets to compare the performance between methods. Later, AF2 was released with the database for Swiss-Prot models. We extracted all available EC functions for UniprotAC identifiers such that we can map them to the AlphaFold models. This gave us enough models to train a method predicting 703 classes. We wanted to test the performance of AlphaFold models compared to PDB structures and created PDB300. Later we decided to merge all the data to get the most training samples at relatively good quality.

Price, CSA, and PrOSPECCT are datasets we obtained from literature; these consist of harder cases covering many facets of protein structures. Since these datasets are designed for proteins, we had to filter non-enzymes and retrain our networks such that we have no samples with 30% sequence identity to the training set. We chose Price as it consists of proteins that have been misclassified based on sequence methods. CSA consists of catalytic sites, making it perfect for investigating if our networks pick up on the catalytic mechanisms in enzymes. PrOSPECCTs is a dataset designed for improving binding site prediction tools, covering many facets of binding sites, ideal for testing the localized 3D descriptor.

REVIEWER 2:

What is TopEC, really? It's a relatively easy issue to address but a major oversight, in my opinion. The first mention of TopEC is in the title, abstract and introduction. Then, for 10 pages of the manuscript the name is gone till page 14. Instead, SchNet and DimeNet++ are mentioned prominently in the beginning of the Results. Clearly, TopEC uses SchNet or(?) /and(?) DimeNet++ networks as a part of the algorithm, but it's never clearly explained. Please, be more upfront about it.

TOPEC:

We agree that TopEC could be better explained and mentioned more explicitly in the text. TopEC is built on top of the SchNet and DimeNet++ networks for the graph convolutional layers. We further explained the TopEC framework and replaced most of the mentions of SchNet and DimeNet++ with TopEC-distances and TopEC-distances+angles. These changes have been introduced between lines 161 and 281 and are marked in green in the manuscript.

REVIEWER 2:

What would TopEC produce if we give it non-enzymes? Do the users need to know a priori if the query protein is an enzyme?

TopEC:

If the user gives TopEC a non-enzyme, it would predict an enzyme class, similar to what is required for CLEAN. We see this as an advantage as this allows the user to scan a protein for enzymatic activity even if the protein is not thought of as an enzyme. While some proteins are not classified as enzymes, they display enzymatic activity, e.g., ABC Transporters hydrolyzing ATP or GTP hydrolysis by G proteins. If we train with a single separate class for non-enzymes, proteins like these can influence the reliability of the prediction. Even if they are marked as non-enzymatic in protein databases, enzymatic activity might enter the dataset for this class. We added the following to the introduction:

TopEC is trained without prior knowledge if a protein is an enzyme or not. This allows the user to scan proteins not classified as enzymes for enzymatic activity. For example, ABC transporters hydrolyze ATP²⁷ and G proteins hydrolyze GTP²⁸.

REVIEWER 2:

336-337 “Generally, we predict an enzyme class either completely correct or wrong, leading to the hourglass-shaped AUPR curves.” This is a major observation that lacks discussion it deserves.

TopEC:

This is a consequence of reconstructing our datasets to obtain a fair performance on PrOSPECCTs. Some enzyme classes have a few distinct folds available. If we remove the enzymes with >30% sequence similarity to the test set, we are often left with too few training examples to learn from, leading to the broad AUPR at 0. We rewrote the section to read (lines 375-378):

Generally, for PrOSPECCTs, we predict an enzyme class either completely correct or wrong, leading to the hourglass-shaped AUPR curves. For the wrongly predicted classes, we lack the diversity in ECs for distinct folds after removing any training samples with > 30% sequence similarity to PrOSPECCTs.

REVIEWER 2:

457-462 Could the authors explain in more detail why they decided to analyze explainability based on a much less-performing model (apart from technical issues)? To shed more light on the way the model works (and hence reduce the methodological bias), it would be fair to analyze also some negative cases (i.e. when models gave incorrect predictions).

TopEC:

While the atom-based model performs worse in predicting enzyme function, the chemical function is more accurately represented by a single node for each atom. We wanted to investigate the differences in explainability based on this difference between residues and atoms. If the model could pinpoint specific atoms, we could use this information to better understand the chemical function. We analyzed the 90 wrongly predicted cases of the CSA dataset for atom resolution and added the explainability results to the SI. While we see higher importance for side-chain atoms compared to

backbone atoms, it is not as prominent on the interacting atoms as in the correctly predicted results. We added the following in lines 576-579:

We also obtained the importance for each catalytic and binding residue for all cases wrongly predicted at atomistic resolution (Figure S19). The importance of side-chain atoms is generally higher than for backbone atoms. However, the importance is less prominent for the interacting atoms than in the correctly predicted results.

REVIEWER 2:

164 The term “fold split” used throughout this manuscript is somewhat misleading. First, because the authors in fact perform a sequence-based clustering at 30% sequence identity, which is a sequence-based and not a “fold” split. Since for all evaluated datasets 3D structures exist (either experimental or models), a real fold split should rely on structures not sequences. Second, the datasets used are on the orders of 10,000s entries - MMSeqs2 is fast but not accurate at low sequence identity values, so using a more exact method (e.g. CD-HIT) seems to be feasible on those datasets.

TopEC:

We agree that using a program such as FoldSeek¹ would have been better to obtain a fold-split but was not available at the time of work. From our understanding, CD-HIT is more exact but does not provide clustering under 40% identity due to the lower performance. They provide an alternate script for clustering at lower identities, PSI-CD-HIT, using BLAST to calculate the similarities². However, according to the original MMSeqs2 paper³, MMSeqs2 profile clustering has a similar performance to PSI-BLAST. Hence, we chose to use MMSeqs2 for the speed advantage.

While the split is based on sequence, we call it a “fold” split as it is often used as a term in enzyme function prediction papers and 30% is a widely accepted number for this split. At 30% sequence identity, the overall fold of a protein could be different from other proteins. Sometimes, this is also called the “Twilight zone” in older literature, as it has been shown to be between 25-40% depending on the system.

We added more citations when we explain the fold bias and added the following sentence in lines 144-146:

Typically, fold bias is removed by clustering the training, validation, and test splits by 30% sequence similarity^{33,34}. Hence, we call our split “fold split”.

¹<https://www.nature.com/articles/s41587-023-01773-0>

²<https://academic.oup.com/bioinformatics/article/26/5/680/212234>

³<https://www.nature.com/articles/nbt.3988>

REVIEWER 2:

458-460 It's unclear if the drop in the number of correct predictions stems only from the fact that the method requires more computing or are there any other factors at play? Did TopEC just fail to compute predictions in 90 cases where all-atom is worse? Or do the authors just speculate that it's worse because we're not able to use as many atoms in the descriptor? One possible experiment would be to limit the Calpha descriptor to the same atoms which the all-atom model could compute, then compare. I would like to see some evidence for one or the other.

TopEC:

Thank you for your comment. This was indeed speculation. We tested all 90 cases with the C α descriptor limited to the same atoms the all-atom model could use in computations. None of the 90 cases were correctly predicted with these descriptors. We added the following (lines 504-507):

We tested the influence of the small binding site representation by predicting these 90 incorrect cases with residue resolution limited to the positions in the atom resolution. For all 90, we did not obtain correct predictions with this residue resolution model.

REVIEWER 2:

592-593 As far as I am concerned, all databases, save for NMR data, have a single conformational state for any given entry, so how is that a flaw of AF2? Also, there is some work showing how to use AF2 to predict multiple conformational states (e.g. <https://www.nature.com/articles/s41586-023-06832-9> or <https://www.sciencedirect.com/science/article/pii/S0959440X23001197>). Please, clarify.

TopEC:

This is not at all a flaw of AF2 or other databases. It remains ambiguous if the state of the model is based on, e.g., a specific allosteric state, or the apo- or holo-state of the enzyme. For the structures obtained from the PDB, one could (sometimes) infer this information from the crystallized ligands and literature. To learn enzymatic function and interactions from local interactions, it could be that multiple states of the enzyme can be beneficial. We certainly agree that methods such as AlphaFlow¹ or MSA tuning might be able to convey this information to the system and can be an interesting research avenue for further research. We added the following to the discussion (lines 660-662):

Recent methods such as AlphaFlow or MSA tuning would allow for the creation of databases with multiple conformations for an enzyme based on various states of proteins. Training methods on such databases might improve the predictive quality.

¹<https://arxiv.org/abs/2402.04845>

MINOR COMMENTS

We grouped the following comments as minor comments requiring rewording or altering small mistakes directly in the manuscript. We thank the reviewers for pointing these out.

MC1 31 “molecular function TO enzymes” should be “molecular function OF enzymes”.

TopEC: Changed to “of enzymes”.

MC2 40-41 “significantly improved EC classification” - the sentence doesn't say with regards to what the classification is improved.

TopEC: Changed to “significantly improved EC classification compared to 2D GNNs”.

MC3 63-67 How does the statement “often molecular function cannot be deduced directly from the structural representation, or the enzyme sequence is annotated incorrectly in databases” relate to the subsequent sentence i.e. how computational methods overcome these experimental limitations?

TopEC: We reworded this to be clearer. It now reads:

Determining enzyme function experimentally for many sequences is time-consuming, often enzyme function cannot be deduced directly from the structural representation, or the wrong enzyme function has been annotated to the sequence in databases⁸. Computational methods using enzyme structures as input can close this gap and allow for high-throughput enzyme function prediction.

MC4 68-70 Among examples of the use of GNN, function prediction seems to be missing. deepFRI (ref. 25) is one such example, against which TopEC is even benchmarked.

TopEC: We added this to the examples.

MC5 97 “F1-score of 0.71”... in the abstract (line 41) the reported value is 0.72. It’s also unclear where this number comes from. Table 1 never reports a value of 0.71 or 0.72

TopEC: We apologize for this typing error for 0.72. This value is obtained from the combination of computational and experimental enzyme structures in Figure 2. We explicitly mention this in the text now.

MC6 Fig. 1a abbreviations RBF and SBF are not explained in the figure caption. They’re only explained towards the end of the manuscript in lines 749-751.

TopEC: We have written out the abbreviations in the figure caption.

MC7 Fig. 1a overlaying DimeNet++ and SchNet architectures is space-efficient but I find it hard to follow what exactly are the differences between them. Presenting the architectures on top of each other could improve legibility.

TopEC: We chose to overlay the architecture to show the many commonalities. We tried presenting the architectures on top of each other but then the image becomes too large if we want to maintain legibility. We added a singular version to the SI (Figure S1).

MC8 112-113 “Residues colored in blue are selected for the input, whereas those in red are discarded.” this sentence is a part of the caption for (b) but I believe it refers to (c) and (d). Otherwise, it makes little sense to me why Ser would be considered and Thr discarded.

TopEC: This was a mistake from re-ordering the panels. We corrected the captions.

MC9 119 What is “fold bias”? the term is used in the abstract and here for the first time. It seems that the authors have a clear intention.

TOPEC: We added a sentence explaining this at the start of the Results section (lines 140-144).

For example, TIM barrels and Rossmann folds are groups with similar supersecondary structures but catalyze many different reactions. If we solely took the overall shape (fold) into account, we would neglect the minor differences leading to different functions. We call this the fold bias.

MC10 132 The way point (1) is phrased is difficult to follow. The authors intend this to be a positive thing but it sounds negative.

TOPEC: Thank you for pointing this out. It is intended to sound positive. We altered the text to sound more positive. It now reads:

1) We focus the network attention to learn the enzyme function from the binding site representation of the protein.

MC11 147 Data S1 and Data S2 are on GitHub but their location is not mentioned in the text.

TOPEC: We plan to add a permanent DOI for all supplementary data and datasets upon acceptance of the paper to the data availability section. We expect the supplementary data to also be available through the publisher as a separate section for Data S1 and Data S2. Furthermore, we also wrote in the data availability section that supplementary data is also available on the git repository.

MC12 161 and other places - BindingMOAD is inconsistently named. Sometimes it's referred to as Binding MOAD and sometimes BindingMOAD.

TOPEC: We changed all occurrences to Binding MOAD.

MC13 164 add “For the fold split” we used MMSeqs2.

TOPEC: We changed the sentence.

MC14 164 “cluster out database BY 30%” should be “cluster our database AT 30%”.

TOPEC: We changed the sentence.

MC15 187-188 “Structure-based predictions...” this sentence seems out of place here. Please, either clarify or remove.

TOPEC: We removed the sentence

MC16 189 + also Fig. 2D what are the DS1-DS7 categories? Later on it's alluded to (e.g. lines 338-339, 349) but it's never explained in detail.

TOPEC: This is the name given to the sections by the original author. We added a reference in the manuscript and added a table to the SI (Table S1) explaining the datasets in more detail.

MC17 Table 1C why only 2 decimal places while in all other sections of Table 1 there are 3 decimal places reported?

TOPEC: We have rounded all values to two decimals.

MC19 220-221 Please, substantiate the claim by providing a suitable reference. Also, the sentence seems to stand in contradiction to eg. 132-133, 241-243, 257-258.

TOPEC: We substantiated the claim by adding a reference. However, both can be true. We see that datasets with higher fold overlap perform better and that directing towards the binding center improves the performance.

MC20 221-223 Sizeably better performance for temporal split as compared to fold split in the case of TopEC models holds only for BindingMOAD and combined datasets but is less evident for the TopEnzyme dataset. Why?

TOPEC: When we investigated the data in TopEnzyme [<https://doi.org/10.1093/bioinformatics/btad116>] the models are of good quality. We think the difference in performance is due to having too few samples to learn enzyme function in both the case of temporal and fold split.

MC20 Moreover, such discrepancy is not observed for DeepFRI. Does it mean that DimeNet++ and SchNet are more dependent on structural homology than other methods, e.g., DeepFRI?

TOPEC: Yes, DimeNet++ and SchNet represent the 3D structure more explicitly, thus overlap between the fold information leads to a higher performance. In DeepFRI the contact map representation is a projection from 3D to 2D topology leading to information loss of the topological structure.

MC21 Fig. 2 Labels (a)-(d) are a part of the title of each plot and thus are poorly legible.

TOPEC: We changed the labels.

MC22 413 Would it be possible to relate residue translation to experimental resolution?

TOPEC: The residue translation can be related to experimental resolution. Roughly, the uncertainty of coordinates \cong crystal resolution / 6¹. So, a translation of 0.5 Å, when the performance is already impacted, would be akin to a crystal resolution of 3 Å. This is also in line with the crystallography “rule of thumb”, where proteins at crystal resolution of <1 Å are considered high resolution, but at 3 Å we must infer parts of the atomic structure.

¹<https://pubmed.ncbi.nlm.nih.gov/12203463/>

“rule of thumb”: <https://pdb101.rcsb.org/learn/guide-to-understanding-pdb-data/resolution>

MC23 Fig. 3g presented 3D structures are too small to appreciate the insights that the authors arrive at.

TOPEC: We added larger figures to the SI (Figures S10-14)

MC24 464-527 Paragraphs provide an eloquent description of what may be observed from TopEC results but are those insights correct? I am lacking some (more) references to the literature that would substantiate the observations made on the basis of TopEC predictions.

TOPEC: These insights are matched to the insights obtained from the original literature for each system. To make this clearer, we repeated the citations for the observations matched to the literature.

MC25 557-558 The statement may be true (compare comments to lines 458-460) but also the authors repeatedly claim that the method does not scale.

Using the residue resolution obtains the best performance, however, atom resolution performance is not much worse (Table 2). The method is hard to scale for atomistic graphs during the training regime, however, we still allow the user to make predictions without the need for abstraction of the 3D structure.

MC26 577-579 I read through the whole paper at this point but I am still not sure if TopEC does it automatically out-of-the-box or are the users deferred to 3rd-party tools to predict binding sites?

TOPEC: TopEC does not predict binding sites out-of-the-box. Users might have different preferences. To accommodate users, we added a section to the repository detailing how to obtain binding sites with P2Rank and added our scripts in GitHub for obtaining the binding site information.

MC27 614-615 The authors say that GNNExplainer doesn't really work since it's designed for 2D GNNs and then go on in lines 616-633 to discuss its results. I find a lengthy discussion of a 3rd party tool which according to the authors “might not generalize as well to 3D GNNs” out of place here.

TOPEC: We shortened the discussion to the most important points of GNNExplainer (lines 677-683).

Reviewer: 3

TOPEC:

Thank you for your feedback and for taking the time to co-review the manuscript.

Response to the reviewer's 2nd comments

We are very grateful to the reviewer for pointing out important points, which we have addressed as detailed below. Particularly, we recognize their commitment to comprehensive comparisons of academic work, with a view to enhancing the longevity and relevance of published papers. We have prepended each comment and answer with **REVIEWER #** and **TopEC** to mark comments and responses.

REVIEWER 2:

[258-265] I encourage the authors to conduct a comparative analysis to illustrate how 30% sequence identity translates into Foldseek's structural clustering. Ideally, a small test could be added to show how TopEC performs on folds derived using Foldseek. Despite Authors' clarification that the tool was not available at the time of work, Foldseek has been available as a full Nature Biotechnology paper for over a year now (10 months at the time of 1st review) and the tool has been widely used by the community for 2.5 years already (since Feb 2022).

TopEC:

We used Foldseek with the same common settings as for MMSeqs2, except for the sequence identity cut-off, which we only used for MMSeqs2. We calculated the Jaccard similarity based on a pairwise comparison between clusters from FoldSeek or MMSeqs2 to get an initial idea of the cluster similarities. In detail, for each cluster pair, we obtained all member pairs and calculated the intersection and union between all pairs to obtain the Jaccard similarity. We found 76 M total pairs (union) and 32 M shared pairs (intersection), which resulted in a Jaccard similarity of 0.43. Thus, both methods provide distinct clusterings that more often than not are different.

Afterwards, we tested our biggest model (TopEC trained on AF2 and PDB structures) to compare scores obtained for both clustering methods. For the MMSeqs2 test set, the results had been reported in Figure2 already. For the Foldseek test set, we removed all clusters with members present in the validation and training set and were left with 2666 samples in the original test set that have no common fold in the training or validation set according to Foldseek. Note that we did not retrain the network on a training and validation set following Foldseek criteria. In contrast, we limit the performance assessment to the enzymes that are clustered according to Foldseek.

The results indicate that using a 30% sequence identity cut-off may not completely eliminate fold bias, as we obtained F1-scores of 0.69 when classifying the main class, 0.61 for the subclass, 0.57 for the sub-subclass, and 0.52 for the designation classification. We thus added the following text to the end of the chapter discussing the performance on large datasets:

[369-378] To further evaluate the impact of the fold bias on the network's performance, we generated an additional test set using Foldseek clustering. For initially comparing the Foldseek clustering to the MMSeqs2-based clusters, the Jaccard similarity was calculated for each enzyme pair present in the clusters. The similarity is 0.43, indicating how the clusters formed by Foldseek are distinct from the MMSeqs2 clusters. After the removal of all Foldseek clusters with overlap with the training and validation sets from the original test set (Figure 2), the F-score is 0.69 for the main class, 0.61 for the subclass, 0.57 for the sub-subclass, and 0.52 for the designation classification. Note that the network was not retrained on a training and validation set following Foldseek criteria but the performance assessment was limited to the enzymes that are clustered according to Foldseek. The result suggests that a 30% sequence identity cut-off may not fully eliminate fold bias.

REVIEWER 2:

[513-519] Providing an explainability analysis for the Ca descriptor that leads to more accurate predictions as compared to an all-atom based approach would give valuable insights into how the method operates — specifically, which backbone atoms are utilized in the inference step. Additionally, the authors could discuss whether the interacting atoms identified by this analysis hold biological significance.

TOPIC:

We have clarified the analysis of why the Ca descriptor leads to more accurate predictions and clarified that we only use the Ca position of the backbone for inference. Additionally, we have expanded the section to better describe our investigation of biologically significant atoms and how the GNNExplainer identifies atoms with high-importance values. The revised section now reads as follows, with the major additions highlighted in yellow:

[520-548] When using the C α position in the local descriptor, information on the position of side chain atoms is lost, as we only encode the C α position of the backbone labeled with the amino acid type. Our analysis revealed that specific C α backbone positions were identified as critical (Figure 3g). However, these positions appeared to alternate without a clear pattern. The alternation does not coincide with the direction of residues in α -helices and β -sheets that point toward the binding site. We could not link the identified positions to known chemical or biological expertise. This suggests for the C α position in the local descriptor that the model may rely on structural features not easily interpretable with established biochemical knowledge. Alternatively, the patterns might result from GNNExplainer highlighting noisy patterns that arise from the complex interplay of graph features in the network.

To investigate this further, we adapted GNNExplainer to investigate a TopEC network trained on AlphaFold2 structures with a fold split and the closest 150 heavy atoms to the binding center with the same CSA dataset as in the previous section. The performance decreases from 232 to 142 correct predictions when using the higher-resolution atomistic descriptor. This drop likely reflects the increased complexity of the graph representation, which could obscure the signal for enzymatic function already encoded in the local 3D position of the amino acid chain. For example, CLEAN achieves impressive performance using only sequence information and the ESM base model. Adding more atoms to the graph may introduce noise, making it harder for the model to isolate functionally relevant features. We tested the influence of the small binding site representation by predicting these 90 incorrect cases with residue resolution limited to the positions in the atom resolution. For all 90, we did not obtain correct predictions with this residue resolution model.

However, we do obtain interesting insights into the network's local properties. For correctly predicted enzymes, we found that biologically relevant atoms, such as those participating in educt stabilization and the catalytic reaction tended to have high importance values. We exemplified this for three serine endopeptidases at residue (Figure 4a-c: left) and atomic resolution (Figure 4a-c: right), repressor LexA62, type 1 signal peptidase63, and GlpG64.

REVIEWER 2:

[595-598] Based on the statistics and conclusions drawn in this section, is it feasible to add a refinement step by assigning greater importance to certain interacting atoms, potentially improving the accuracy of functional predictions?

TopEC:

This is an interesting direction to research; while assigning greater importance to certain interacting atoms might improve the accuracy of the functional prediction, it needs to be tested if this will hold true as the $C\alpha$ descriptor networks still outperforms the full atomistic descriptor.

We do think that the atomistic descriptors will help with the explainability of the networks. If we train both an atomistic and $C\alpha$ network simultaneously, we can make a consensus prediction and obtain importance values for the atomistic graphs with the predictive performance of the $C\alpha$ networks. Additionally, we can add a refinement step with a learning objective of matching the importance of atoms participating in catalysis obtained from experimental data (e.g. Catalytic Site Atlas). This could match the importance values more closely for structures for which we have no experimental data available. This improved importance could then be used for finding and designing enzymes with specific catalytic activity.

However, we think that the required architectural change to the model is outside the scope of the current research. Instead, we provide the idea to the community and added the following sentence after 595-598:

[619-624] Using the importance results for a refinement step with a learning objective of matching the importance of atoms participating in catalysis according to experiment may improve the predictive performance but this remains to be tested. This might lead to a closer match of importance prediction and function prediction performance, which might be useful for finding and designing new enzymes and deciphering how enzyme function prediction networks learn.

Response to the reviewer's 3rd comments

We are very grateful to the reviewer for pointing out important points, which we have addressed as detailed below. We have prepended each comment and answer with **REVIEWER #** and **ToPEC** to mark comments and responses.

REVIEWER 2:

I thank and applaud the authors for addressing all of the concerns well!

ToPEC:

Thank you for the positive feedback.

REVIEWER 2:

The repository is generally well-structured, exhaustive and contains an useful README. Some minor remarks about the GitHub repo:

1. following the installation instructions provided in the README file does not work without fixing. The `requirements.txt` file seems to be not specified correctly (some conflicts exist) and some dependencies also seem to be missing (e.g. Biopython which is required by some of the modules). I was able to make it work with some minor tinkering but could be easily fixed by testing on a clean environment and/or including channels in the specification + not over-specifying some software versions (e.g. I wasn't able to install with `torchvision==0.16.2+cu121` which assumes CUDA 11.2)

ToPEC:

We removed the strict requirements to let the pip solver do its work. This solved the issues on our side. We suggest opening up a pull request with one's versioning should the issues remain on one's side. Furthermore, the list of packages used is mentioned for one's own setup of the python environment.

REVIEWER 2:

2. some Python scripts in the main folder are not described in the readme, e.g. `prepare_data.py` or `run_dataset_create.py`

ToPEC:

One of the scripts was a legacy file that was used for an older version of how we prepared the dataset. The other script has been described in the README.md file now.

REVIEWER 3:

The code appears complete and professionally organized, with clear and detailed instructions for retraining the model and making inferences. However, I encountered dependency conflicts while attempting to run the code during the creation of the virtual or Conda environment. I recommend that the authors enhance the reproducibility of the Python environment. This could involve upgrading the Python version from 3.9 to a more recent one or providing a YAML file as an alternative to the requirements.txt file.

TopEC:

We relaxed the strict requirements for the “requirements.txt” file. We do not use the conda environments as we are limited with which conda channels we are allowed to access at our research institute. If one wants to install with conda, we would suggest to install the packages manually in one’s environment. Just before the pip installation section in the readme are all the required Python packages to run the code.

In this paper, van der Weg et al. present TopEC, a graph neural network for predicting Enzyme Commission classes based on three-dimensional enzyme structures. The proposed network is based on modern SchNet and DimeNet++ architectures, allowing it to learn from complex relationships between biochemical features and shape-related characteristics of enzyme structures. The resulting models show impressive performance under certain circumstances and present exciting opportunities for improving structure-based function classification of enzymes.

Our chief concerns are over the apparent lack of novelty in the proposed method, as well as the paper's difficulty in showing its improvement over the state of the art. Our comments and suggestions are described in greater detail below.

1. Please elaborate more on the contribution of the paper. A significant component appears to be the combination of pre-existing models; however, that in itself is of limited interest. If the authors made modifications to the models or introduced new techniques of their own, a more explicit description would help establish the novelty of their work.
2. Please show more clearly how the model outperforms the current models (EnzyNet and DeepFRI) and how it compares for the larger datasets (PDB). Specifically, please include additional performance comparison with EnzyNet on the PDB dataset, which was reported to be 78.4% in previous literature.
3. The authors note that few methods use explicitly encoded 3D information for predicting EC numbers. What are the reasons for such scarcity? Elaborating more on this point will help establish the importance of the problem being addressed.
4. Performance on the Price dataset seems to leave more questions than answers. This is indeed a difficult test case, but to what extent and in what ways? How does the proposed model compare to other methods?
5. Regarding the ProSPECCTs dataset, the contrast between poor accuracy on DS1 and good accuracy on DS5 and DS6 begs more explanation. Both pertain to structures with significant structural and ligand similarities, yet the performance is very different. What may be the reasons for this difference?
6. The improvement in understandability seen with the finer-grain atom resolution approach may have been due to the increased focus on the binding site and not necessarily a result of inclusion of additional heavy atoms. Can you comment on that possibility?
7. While there are many potential uses for this method, whether it'll perform well in such diverse applications remains an open question. Please comment why you think TopEC is likely to be an effective approach to the proposed applications, based on the observed results or the intrinsic design of the model.

Some minor concerns

8. Line 82-345: six instances of "subsub-class" or "subsubclass" should be "sub-subclass."
9. Line 303-345: four instances of "mainclass" should be "main class."
10. Line 776: "atom l" should be "atom i."